# Deep learning for quality control of surface physiographic fields using satellite Earth observations

Tom Kimpson[1], Margarita Choulga[2], Matthew Chantry[2], Gianpaolo Balsamo[2], Souhail Boussetta[2], Peter Dueben[2], and Tim Palmer[1]

[1]Department of Physics, University of Oxford, Oxford, UK
[2]Research Department, European Centre for Medium-Range Weather Forecasts (ECMWF), Reading, UK

**Correspondence:** Tom Kimpson (tom.kimpson@physics.ox.ac.uk)

**Abstract.** A purposely built deep learning algorithm for the Verification of Earth-System ParametERisation (VESPER) is used to assess recent upgrades of the global physiographic datasets underpinning the quality of the Integrated Forecasting System (IFS) of the European Centre for Medium-Range Weather Forecasts (ECMWF), which is used both in numerical weather prediction and climate reanalyses. A neural network regression model is trained to learn the mapping between the surface physiographic dataset plus the main meteorologic fields from ERA5, and the MODIS satellite skin temperature observations. Once trained, this tool is applied to rapidly assess the quality of upgrades of the physiographic fields used by land-surface scheme. Upgrades which improve the prediction accuracy of the machine learning tool indicate a reduction of the errors in the surface fields used as input to the surface parametrisation schemes. Conversely, incorrect specifications of the surface fields decrease the accuracy with which VESPER can make predictions. We apply VESPER to assess the accuracy of recent upgrades of the permanent lake and glaciers covers as well as planned upgrades to represent seasonally varying water bodies (i.e. ephemeral lakes). We show that for grid-cells where the lake fields have been updated, the prediction accuracy by VESPER in the land surface temperature (as quantified by the mean absolute error) improves by 0.37 K on average, whilst for the subset of points where the lakes have been completely removed and replaced with bare ground the improvement is 0.83 K. We also show that updates to the glacier cover improve the prediction accuracy by 0.22 K. We highlight how neural networks such as VESPER can assist the research and development of surface parametrizations and their input physiography to better represent Earth's surface coupled processes in weather and climate models.

## 1 Introduction

Accurate knowledge of the global surface physiography, including land, water and ice covers, and their characteristics, strongly determines the quality of surface and near-surface temperature simulations in weather and climate modelling. For instance, water bodies exchange mass and energy with the atmosphere and their thermal inertia strongly influences the lower boundary conditions such as skin temperatures, and surface fluxes of heat and moisture near the surface. Globally, there are $\sim 117$ million lakes - defined as inland water bodies without lateral movement of water - making up around 3.7% of the Earth's land surface (Verpoorter et al., 2014). Their distribution is highly non-uniform, with the majority of lakes located between

$45 - 75°$N in the Boreal and Arctic regions. Lakes are highly important from the perspective of both numerical weather prediction and climate modelling as part of the EC-Earth model. For the latter, lakes generally influence the global carbon cycle as both sinks and sources of greenhouse gases; the majority of lakes are net heterotrophic (i.e. oversaturated with carbon dioxide, $CO_2$), as a result of in lake respiration and so emit carbon into the atmosphere (Pace and Prairie, 2005; Tranvik et al., 2009). Total $CO_2$ emission from lakes is estimated at $1.25 - 2.30$ Pg of $CO_2$-equivalents annually (DelSontro et al., 2018), nearly $20\%$ of global $CO_2$ fossil fuel emissions, whilst lakes account for 9-24 % of $CH_4$ emissions, the second largest natural source after wetlands (Saunois et al., 2020). These rates of greenhouse gas emission are expected to rise further if the eutrophication (i.e. nutrient concentration increase) of the Earth's lentic systems continues. With regards to weather, freezing and melting of the lake surface modifies the radiative and conductive properties and consequently affects the heat (latent, sensible) exchange and surface energy balance (Franz et al., 2018; Huang et al., 2019; Lu et al., 2020). Considering particular examples, over Lake Victoria convective activity is suppressed during the day and peaks at night, leading to intense, hazardous thunderstorms (Thiery et al., 2015, 2017); Lake Ladoga can generate low level clouds which can cause variability in the 2m temperature of up to 10 K (Eerola et al., 2014); the Laurentian Great Lakes can cause intense winter snow storms (Notaro et al., 2013; Vavrus et al., 2013). Moreover, as a result of the increased temperatures due to climate change, lakes become more numerous due to the melting of glaciers and permafrost. Additionally, the higher temperatures mean that previously permanent lake bodies become seasonal or intermittent. There is then evidently a huge potential return in the ability to accurately model the location, morphology and properties of lakes in weather and climate models.

The Integrated Forecasting System (IFS) at the European Centre for Medium Range Weather Forecasts (ECMWF) is used operationally for numerical weather prediction and climate modelling. Earth-system modelling in the IFS can be broadly categorised into large-scale and small-scale processes. Large-scale processes can be described by numerically solving the relevant set of differential equations, to determine e.g. the general circulation of atmosphere. Conversely, small-scale processes such as clouds or land-surface processes are represented via parametrisation. Accurate parametrisations are essential for the overall accuracy of the model. For example, the parametrisation of the land surface determines the sensible and latent heat fluxes, providing the lower boundary conditions for the equations of enthalpy and moisture in the atmosphere (Viterbo, 2002).

Lakes are incorporated in Earth-system models via parametrisation. At ECMWF the representation of lakes via parametrisation was first handled by introducing the Fresh water Lake model FLake (Mironov, 2008) into the IFS. FLake treats all resolved inland waterbodies (i.e. lakes, reservoirs, rivers which are dominating in a grid-cell) and unresolved or sub-grid water (i.e. small inland waterbodies and sea/ocean coastal waters which are present but not dominating in a grid-cell). Its main input fields are lake location and lake mean depth. The broad impact of the FLake model (i.e. areas where it is active) and the important role that waterbodies play in human life can be illustrated by analysing ECMWF fields of the fractional land sea mask and the inland waterbody cover alongside the population density field (i.e. inhabitants per $km^2$) based on the population count for 2015 from the Global Human Settlement Layers (GHSL), Population Grid 1975-2030 (Freire et al., 2016; Schiavina et al., 2022) at 9 km horizontal resolution.

Globally FLake is active over 11.1% of the grid-cells; considering only land grid-cells, then FLake is active over 32.4% of the points. According to the population data, 64.5% of densely populated areas (at least 300 inhabitants per km$^2$) are situated within a 9 km radius of a permanent waterbody (i.e. inland water or sea/ocean coast), with 31.2% being in the vicinity of at least 1 km$^2$ waterbody - emphasising how essential waterbodies are in human life. In some regions this role may be even more crucial than in the others. For example in North America 45.7% of the densely populated areas are close to a 1 km$^2$ waterbody; in Australia where only 0.5 % of the land is populated, two thirds of the population live within 9 km radius of a permanent waterbody of at least 1 km$^2$, with the majority of people living on the ocean coast.

It is a continuous enterprise to update the lake-related physiographic fields used as input by land-surface scheme to better represent small-scale surface processes. It is however challenging to do it accurately as the majority of lakes which are resolved at a 9km grid spacing have not had their morphology accurately measured, let alone monitored, whilst 28.9% of land and coastal cells are treated for sub-grid (i.e. covering half or less of a grid cell) water. When introducing an updated lake representation it is difficult apriori to determine the additional value gained through doing so. There are two key factors here:

– Are the updated fields closer to reality?

– Do the updated fields increase the accuracy of the model predictions?

The first point is straightforward; we want our fields to better represent reality. If the lake depth of some lake is updated from 10m to 100m we want to be sure that 100m is closer to the true depth of the lake. For the second point, even if the updated fields are accurate, are they informative in the sense that they enable us to make more accurate predictions? For instance, the main target of lake parametrization is to reproduce lake surface water temperatures (and therefore evaporation rates). If lake parametrisation input fields are updated to better represent different types of inland waterbodies, the time variability of inland waterbodies and/or the lake morphology fields use more in situ measurements, does this additional information allow for more accurate predictions of the lake surface water temperatures? Is it therefore worthwhile to spend several person-months to update/create a lake-related field? Since the resulting updated fields are ultimately used operationally, it is essential to ensure the accuracy of the fields and prevent any potential degradation or instability of the model. This problem of quickly and automatically checking the accuracy and information gain of updated lake-related fields is the aim of this work.

Numerical weather prediction and climate modelling are domains that are inherently linked with large datasets and complex, non-linear interactions. It is therefore an area that is particularly well placed to benefit from the deployment of machine learning algorithms. At ECMWF, advanced machine learning techniques have been used for parametrisation emulation via neural networks (Chantry et al., 2021), 4D-Var data assimilation (Hatfield et al., 2021) and the post-processing of ensemble predictions (Hewson and Pillosu, 2021). Indeed, the early successes of these machine learning methods have led to the development of a 10-year roadmap for machine learning at ECMWF (Düben et al., 2021), with machine learning methods looking to be integrated into the operational workflow and machine learning demands considered in the procurement of HPC facilities. The ongoing development of novel computer architectures (e.g. GPU, IPU, FGPA) motivates utilizing algorithms and techniques

which can efficiently take advantage of these new chips and gain significant performance returns. In this work we will demonstrate a new technique for the Verification of Earth-System ParametERisation (VESPER) based on a deep learning neural network regression model. This tool enables the accuracy of an updated water body-related field to be rapidly and automatically assessed, and the added value that such updated fields bring to be quantitatively evaluated.

This paper is organized as follows. In Section 2 we describe the construction of the VESPER tool - the raw input data, the processing steps and the construction of a neural network regressor. In Section 3 we then deploy VESPER to investigate and evaluate updated lake-related fields. Discussion and concluding remarks are made in Sections 4 and 5 respectively.

## 2 Constructing VESPER

In order to rapidly assess the accuracy of new surface physiography fields and if their use in the model increase the accuracy with which we can make predictions, a neural network regression model (VESPER, hereafter) that can learn the mapping between a set of input features $x$ and targets $y$ is constructed. In this case the features are the atmospheric and surface model fields, such as 2 metre temperature from ERA5 reanalysis, and the surface physiographic fields, such as orography and vegetation cover used to produce ERA5 reanalysis. See Table 1 for the full list of variables used. The target is the satellite land surface temperature (LST; skin temperature from MODIS Aqua Day MYD11A1 v006 collection). Once trained, VESPER can then make predictions about the skin temperature given a set of input variables, i.e. atmospheric and surface model fields, and surface physiographic fields. In turn, these predictions can then be compared against observations, i.e. satellite skin temperature, and VESPER's accuracy evaluated. By varying the number, type and values of the input features to VESPER and observing how the accuracy of its predictions change, some conclusions on if and how features can increase predictability of an actual atmospheric model can be drawn. Moreover, by isolating geographic regions where the predictions get worse with new/updated surface physiographic fields, areas where these fields might be erroneous or not informative enough can be identified. Due to the inherent stochasticity of training a neural network regression model it is also possible for different models to settle in different local minimums i.e. the network variance/noise. To understand the significance of this, every VESPER configuration was trained four times, each time with a different random seed.

In this section we will now describe the data used for the features $x$ and targets $y$ in the neural network regression model, how various data types are joined together, and the details of VESPER's construction.

### 2.1 Features and targets

VESPER's input feature selection (see Table 1) followed (i) permutation importance results for atmospheric and surface model fields - only fields with the highest importance were chosen; and (ii) expert choice for surface physiographic fields. As a first attempt it was decided to test the current methodology for lake related information, therefore fields that could be most affected by the presence or absence of water were selected, e.g. if lake had to be removed then some other surface had to appear, such

as bare ground, high or low vegetation, glacier or even ocean. Moreover the surface elevation had to change. Changes to the orographic fields will have important influences on temperature through e.g. wind, solar heating, etc. Lake depth changes are similarly important, influencing how a lake freezes, thaws, mixes and its overall dynamical range (i.e. changes of temperature and mixed layer depth). VESPER's target selection followed globally available criteria and the satellite LST is quite well observed globally and with high temporal pattern, daily or even several times a day depending on the location.

## 2.2 Data sources

There are three main sources of data. The first is selection of surface physiographic fields from ERA5 (Hersbach et al., 2020) and their updated versions (Choulga et al., 2019; Boussetta et al., 2021; Muñoz Sabater et al., 2021a) used as VESPER's features. As a shorthand we will refer to the original ERA5 physiographic fields as version "V15" and the updated versions as "V20". The second is a selection of atmospheric and surface model fields from ERA5, also used as VESPER's features. The third is day-time LST measurements from the Moderate Resolution Imaging Spectroradiometer (MODIS) onboard the Aqua satellite (GSFC), used as VESPER's target variable.

### 2.2.1 Surface physiographic fields

Surface physiographic fields have gridded information of the Earth's surface properties (e.g. land-use, vegetation type and distribution) and represent surface heterogeneity in the ECLand of the IFS. They are used to compute surface turbulent fluxes (of heat, moisture and momentum) and skin temperature over different surfaces (vegetation, bare soil, snow, interception and water) and then to calculate an area-weighted average for the grid-box to couple with the atmosphere. To trigger all different parametrization schemes the ECMWF model uses a sets of physiographic fields, that do not depend on initial condition of each forecast run, or the forecast step. Most fields are constant; surface albedo is specified for 12 months to describe the seasonal cycle. Dependent on the origin, initial data comes at different resolutions and different projections, and is then first converted to a regular latitude-longitude grid (EPSG:4326) at ∼ 1km at Equator resolution, and secondly to a required grid and resolution. Surface physiographic fields used in this work consist of orographic, land, water, vegetation, soil, albedo fields and their difference between initial V15 and updated V20 field sets. See Tables 1 and 2 for the full list of surface physiographic fields and their input sources; for more details see IFS documentation (ECMWF, 2021). As this work is focused on assessing quality of inland water information, main surface physiographic fields are lake cover (derived from land-sea mask) and lake mean depth (see Table 2).

To generate V15 fractional lake cover the GlobCover2009 global map (Bontemps et al., 2011; Arino et al., 2012) is used. This map has a resolution of 300m, corresponds for the year 2009 and covers latitudes 85°N-60°S; corrections outside these latitudes for the polar regions are included separately. In the Arctic no land is assumed, in the Antarctic data from the high-resolution Radarsat Antarctic Mapping Project digital elevation model version 2 (RAMP2; Liu et al., 2015) is used. To generate V20 fractional lake cover more recent higher resolution datasets and updated methods have been used (Choulga et al., 2019). The main data source is the Joint Research Centre (JRC) the Global Surface Water Explorer (GSWE) dataset (Pekel et al., 2016).

| | |
|---|---|
| Atmospheric and surface model fields (11 fields) | **Pressure**: surface pressure (*sp, Pa*), mean sea level pressure (*msl, Pa*), <br><br>**Wind**: 10 metre U wind component (*10u, m/s*), 10 metre V wind component (*10v, m/s*), <br><br>**Temperature**: 2 metre temperature (*2t, K*), 2 metre dewpoint temperature (*2d, K*), skin temperature (*skt, K*), ice temperature layer 1 (the sea-ice temperature in layer 0-7 cm; *istl1, K*), ice temperature layer 2 (the sea-ice temperature in layer 7-28 cm; *istl2, K*), <br><br>**Surface albedo:** forecast albedo (*fal, 0-1*), <br><br>**Snow:** snow depth (*sd, m* of water equivalent) |
| Main surface physiographic fields (19 fields) | **Orographic fields:** standard deviation of filtered subgrid orography (*sdfor, m*), standard deviation of orography (*sdor, m*), anisotropy of sub-gridscale orography (*isir, -*), angle of sub-gridscale orography (*anor, radians*), slope of sub-gridscale orography (*slor, -*), geopotential (the gravitational potential energy of a unit mass, at a particular location, relative to mean sea level; at the surface of the Earth, this parameter shows the variation in geopotential (height) of the surface, and is referred to as the orography; *z, $m^2 s^{-2}$*), <br><br>**Land fields:** land-sea mask (the proportion of land, as opposed to ocean or inland waters (i.e. lakes, reservoirs, rivers, coastal waters), in a grid-cell; *lsm, 0-1*), glacier mask (the proportion of a grid-cell covered by glacier; *glm, 0-1*), <br><br>**Water fields:** lake cover (the proportion of a grid-cell covered by inland water bodies; *cl, 0-1*), lake total depth (the mean depth of inland water bodies; *dl, m*), <br><br>**Vegetation fields:** low vegetation cover (*cvl, 0-1*), high vegetation cover (*cvh, 0-1*), type of low vegetation (*tvl, -*), type of high vegetation (*tvh, -*), <br><br>**Soil fields:** soil type (*slt, -*), <br><br>**Albedo fields:** UV visible albedo for direct radiation (*aluvp, 0-1*), UV visible albedo for diffuse radiation (*aluvd, 0-1*), near IR albedo for direct radiation (*alnip, 0-1*), near IR albedo for diffuse radiation (*alnid, 0-1*) |
| Additional surface physiographic fields | Difference for all main surface physiographic fields between V15 and V20 field sets, <br><br>Difference between V20 static lake cover and monthly varying lake cover (12 maps in total), <br><br>Saline lake cover (the proportion of a grid-cell covered by saline inland water bodies; units: 0-1) |

**Table 1.** Input features used for training the neural network model VESPER; atmospheric model fields (time varying) were kept the same in all simulations, surface physiographic fields (static) were updated when going from the original data based on GlobeCover2009/GLDBv1 (V15 field set) to GSWE/GLDBv3 (V20 field set); in brackets are variables description (where needed), short name (according to the GRIB parameter database) and units.

| Field category | V15 (initial) | V20 (updated) |
| --- | --- | --- |
| Orographic | SRTM30 Shuttle Radar Topography Mission over 60°N-60°S; GLOBE: Global Land One-km Base Elevation Project data over 90-60°N; RAMP2: high-resolution Radarsat Antarctic Mapping Project Digital Elevation Model version 2 data (Liu et al., 2015) over 60-90°S; BPRC: Byrd Polar Research Center over Greenland; IS 50V: Digital Map Database of Iceland over Iceland | As V15, with corrections of erroneous shift |
| Land | **glm:** GLCC: Global Land Cover Characteristics version 2.0 over 90°N-90°S except Iceland; Icelandic Meteorological Office (IMO) glacier mask 2013 over Iceland<br>**lsm:** GlobCover2009 (Bontemps et al., 2011; Arino et al., 2012) over 85°N-60°S; RAMP2: high-resolution Radarsat Antarctic Mapping Project Digital Elevation Model version 2 data (Liu et al., 2015) over 60-90°S; no land assumed over 90-85°N | **glm:** Norwegian Institute glacier data over Svalbard; Icelandic Meteorological Office (IMO) glacier mask 2017 over Iceland; GIMP: Greenland Ice Mapping Project data (Howat et al., 2014) over Greenland; CryoSat-2 satellite glacier data (Slater et al., 2018) over Antarctica (+ manual gap filling); GLIMS: Global Land Ice Measurements from Space data (GLIMS and NSIDC, 2005, updated 2018) over rest of the globe<br>**lsm:** GSWE: Global Surface Water Explorer (Pekel et al., 2016); glm |
| Water | **cl:** *lsm* (ocean is separated at actual resolution by seeding and removing all connected grid-cells, includes the Caspian Sea, the Azov Sea, The American Great Lakes)<br>**dl:** The Caspian Sea bathymetry; Global Relief Model ETOPO1 (Amante and Eakins, 2009) over the Great Lakes, the Azov Sea; GLDB: Global Lake DataBase version 1 (Kourzeneva et al., 2012) over rest of the globe; 25 meters assumed over missing data grid-cells | **cl:** *lsm* (ocean is separated at 1km resolution by upgraded flooding algorithm following Choulga et al. (2019)<br>**dl:** GEBCO: General Bathymetric Charts of the Ocean (Weatherall et al., 2015) over the Caspian Sea and the Azov Sea; Global Relief Model ETOPO1 (Amante and Eakins, 2009) over the Great Lakes; GLDB: Global Lake DataBase version 3 (Choulga et al., 2014) over rest of the globe; indirect estimates based on geological origin of lakes (Choulga et al., 2014) over missing data grid-cells |
| Vegetation | GLCC: Global Land Cover Characteristics version 1.2. Note that vegetation type represent only dominant type over grid-cell | As V15 |
| Soil | DSMW: FAO/UNESCO Digital Soil Map of the world (FAO, 2003). Note that soil type represent only dominant type over grid-cell | As V15 |
| Albedo | MODIS 5-year climatology (Schaaf et al., 2002); RossThickLiSparseReciprocal BRDF model. Note that Albedo values represent snow free surface albedo | As V15 |

**Table 2.** List of input datasets for the surface physiographic fields for V15 and V20 field sets. V15X and V20X are identical to V15 and V20 respectively, but with the addition of saline lake cover, and monthly varying lake cover fields.

GSWE is a 30m resolution dataset from Landsat 5,7 and 8, providing information on the spatial and temporal variability of surface water on the Earth since March 1984; here only permanent water was used for lake cover generation as it provided a more accurate inland water distribution on the annual basis (Choulga et al., 2019). Differences between V20 and V15 lake cover fields (see Figure 1) are consistent with the latest global and regional information: (i) increase of lake fraction in V20 compared to V15 over northern latitudes is due to permafrost melt leading to a new thermokarst lake emergence, and due to higher resolution input source and its better satellite image recognition methodologies; (ii) reduction of lake fraction in V20 compared to V15 can be explained with several reasons, like anthropogenic land use change (e.g. Aral Sea, which lies across the border between Uzbekistan and Kazakhstan, has been shrinking at an accelerated rate since the 1960s and started to stabilise in 2014 with an area of 7660 km$^2$, 9 times smaller than its size in 1960. GlobCover2009 describes the Aral Sea in 1998, when it was still "only" two times smaller than its 1960 extent, whereas GSWE provides a more up to date map.), use of only permanent water (e.g. Australia, where GlobCover2009 over-represents inland water, as most of these lakes are highly ephemeral, e.g. the endorheic Kati Thanda–Lake Eyre fills only a few times per century. The GSWE updates to this region therefore include only generally permanent water, removing all seasonal and rare ephemeral water.), and change in the ocean and inland water separation algorithm (e.g. north-east of Russia).

To generate V15 lake mean depth (see Figure 2) the Global Lake DataBase version 1 (GLDBv1; Kourzeneva et al., 2012) is used. GLDBv1 has a resolution of 1km and is based on 13000 lakes with in situ lake depth information; outside this dataset all missing data grid-cells (i.e. over ocean and land) have 25 meter value; field aggregation to a coarser resolution is done by averaging. Overestimation of lake depth in summer season can result in strong cold biases and in winter season – lack of ice formation. To generate V20 lake mean depth an updated version GLDBv3 (Choulga et al., 2014) is used. GLDBv3 has the same resolution of ∼1km, but is based on 1500 additional lakes with in situ depth information (in addition to bathymetry information over all Finnish navigable lakes), it introduces distinction between freshwater and saline lakes (this information is currently not used by FLake), and suggests the method to assess the depth for lakes without in situ observations using geological and climate type information; field aggregation to a coarser resolution is done by computing the most occurring value. Verification of GLDBv1 and GLDBv3 lake depths against 353 Finnish lake measurements shows that GLDBv3 exhibits a 52 % bias reduction in mean lake depth values compared to GLDBv1 (Choulga et al., 2019). For a further details on lake distribution and depth, the representation of lakes by ECMWF in general see Choulga et al. (2019) and Boussetta et al. (2021).

To expand V15 and V20 lake description (to V15X and V20X respectively) their salinity and time variability information was generated. Even though static permanent water fits better to describe inland water distribution on average all year round, some areas (in Tropics especially) could benefit from having monthly varying information as they have a very strong seasonal cycle, when size, shape and depth of a lake changes over the course of the year, leading to a significant change in modelling the lake temperature response. Similarly, saline lakes behave very differently to fresh water lakes since increased salt concentrations affect the density, specific heat capacity, thermal conductivity, and turbidity, as well as evaporation rates, ice formation and ultimately the surface temperature. These two properties of time variability and salinity are often related; it is common for

saline lakes to fill and dry out over the course of the season, which naturally also affects the relative saline concentration of the lake itself. To create a monthly varying lake cover first 12 monthly fractional land-sea masks based on JRC Monthly Water History v1.3 maps for 2010-2020 were created. Since the annual lake maps were created taking into account a lot of additional sources the extra condition on the monthly maps that the monthly water is equal or greater than permanent water distribution from fractional land-sea mask is enforced. To create an inland salt lake cover map, the GLDBv3 salt lake list was used. First, in order to identify separate lakes on $\sim$ 1km resolution lake cover (by "lake cover" we refer the maximum lake distribution based on 12 monthly-varying lake covers), small sub-grid lakes and large lake coasts are masked, i.e. grid-cells that have water fraction less than 0.25. Next, the number of connected grid-cells in each lake (i.e. connected with sides only) in the 1-km grid is computed. Then only lakes that have 100 and more connected grid-cells are vectorised [1], as at ERA5 resolution of $\sim$31km the grid-cells are quite large and can include a mixture of freshwater and saline lakes. Finally, saline lake vectors are selected by filtering vectors which have no saline lake point from GLDBv3 located – in total 147 large salt lake vectors, which were further used to filter non-saline lakes at 1km resolution lake cover, finally aggregated to 31km resolution. In the future it is planned to revisit this field and extend the list to include additional data. Note that all non-lake related climate fields such as vegetation cover or orography were updated in V20 field set compared to V15 only in relation to the changing lake fields (i.e. if fraction of lake in the grid cell increased then other fractions like vegetation or bare ground should have decreased accordingly).

### 2.2.2 ERA5

Climate reanalyses combine observations and modelling to provide calculated values of a range of climactic variables over time. ERA5 is the fifth generation reanalysis from ECMWF. It is produced via 4D-Var data assimilation of the IFS cycle 41R2, coupled to a land-surface model (ECLand, Boussetta et al., 2021), which includes lake parametrization by FLake (Mironov, 2008) and an ocean wave model (WAM). The resulting data product provides hourly values of climatic variables across the atmosphere, land and ocean at a resolution of approximately 31km with 137 vertical sigma levels, up to a height of 80km. Additionally, ERA5 provides associated uncertainties of the variables at a reduced 63km resolution via a 10-member Ensemble of Data Assimilations (EDA). In this work ERA5 hourly surface fields at $\sim$ 31km resolution on a reduced Gaussian grid are used. Gaussian grid's spacing between latitude lines is not regular, but lines are symmetrical along the Equator; the number of points along each latitude line defines longitude lines, which start at longitude 0 and are equally spaced along the latitude line. In a reduced Gaussian grid, the number of points on each latitude line is chosen so that the local east-west grid length remains approximately constant for all latitudes (here Gaussian grid is N320, where N is the number of latitude lines between a Pole and the Equator). The main field used from ERA5 is skin temperature (i.e. temperature of the uppermost surface layer, which has no heat capacity and instantaneously responds to changes in surface fluxes) that forms the interface between the soil and the atmosphere. Skin temperature is a theoretical temperature computed by linearizing the surface energy balance equation for each surface type separately, and its feedback on net radiation and ground heat flux is included; for more information see IFS

---

[1]By "vectorised" we mean that we transfer these groups of 100+ grid-cells into a shapefile layer, then we check if inside the shapefile falls a latitude/longitude point from GLDB saline lake list; if yes, this shapefile is kept in the layer, if not – it is deleted from the layer. We apply this shapefile layer as a mask to filter at 1 km resolution fractional grid-cells with lake cover.

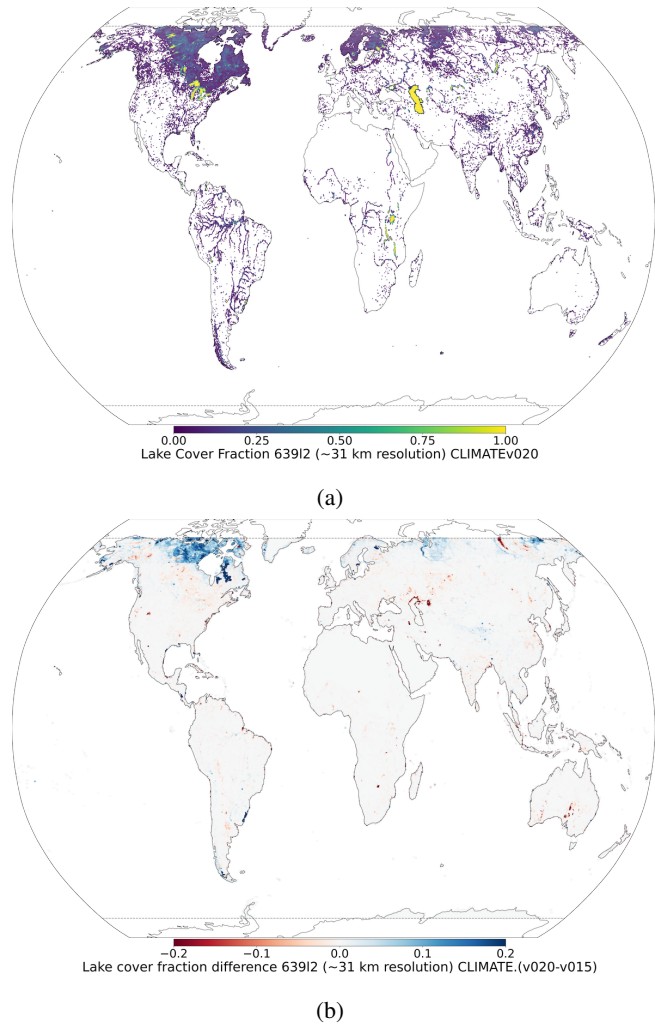

(a)

(b)

**Figure 1.** At ∼ 31km resolution (a) V20 fractional lake cover and (b) difference between V20 and V15 lake covers. Over northern latitudes inland water increase in V20 compared to V15 is due to higher resolution input source and its better satellite image recognition methodologies as well as thawing permafrost; inland water reduction in V20 compared to V15 is due to anthropogenic land use changes (e.g. Aral Sea) or due to use of only permanent water (e.g. Australia) which was proven to better represent inland water distribution on annual basis.

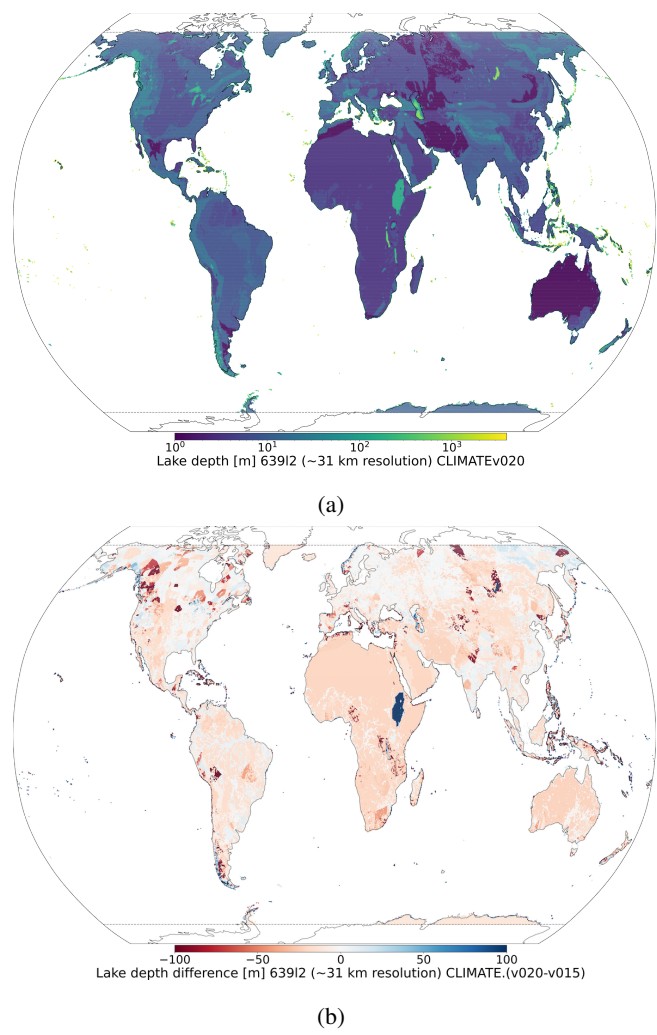

(a)

(b)

**Figure 2.** At ∼ 31km resolution (a) V15 lake mean depth in meters and (b) difference between V20 and V15 lake mean depths. In general lake mean depth has decreased in V20 compared to V15 due to the use of mean depth indirect estimates based on geological and climate information, instead of default 25 meter value over lakes without any information.

Documentation (2021). ERA5 skin temperature verification against MODIS LST ensemble (i.e. all four MODIS observations were used, namely Aqua Day and Night, Terra Day and Night) over 2003-2018 period showed good correlation between two datasets; errors between ERA5 and MODIS LST ensemble are quite small, i.e. spatially and temporally averaged bias is 1.64 K, root-mean square error (RMSE) is 3.96 K, Pearson correlation coefficient is 0.94, and anomaly correlation coefficient is 0.75 (Muñoz Sabater et al., 2021b). ERA5 skin temperature verification against the Satellite Application Facility on Land Surface Analysis (LSA-SAF) product over Iberian Peninsula showed a general underestimation of daytime LST and slightly overestimation at night-time, relating the large daytime cold bias with vegetation cover differences between ERA5 surface physiography fields and the European Space Agency's Climate Change Initiative (ESA-CCI) Land Cover dataset. The use of ESA-CCI low and high vegetation cover over ERA5 has been shown to lead to a complete reduction of the large maximum temperature bias during summer (Johannsen et al., 2019). ERA5 data is obtained via the Copernicus Climate Data Store (CDS; Munoz Sabater, 2019).

### 2.2.3 Aqua-MODIS

Aqua (Parkinson, 2003) is a NASA satellite mission which makes up part of the Earth Observing System (EOS). Operating at an altitude of 700km, with orbital period of 99 minutes, its orbital trajectory passes south to north with an equatorial-crossing times in general of 1.30pm. This post-meridian crossing time has led to it sometimes being denoted as EOS PM. Launched in 2002 with an initial expected mission duration of 6 years, Aqua has far exceeded its initial brief and until recently has been transmitting information from 4 of the 6 observation instruments on board. Here we use information only from MODIS instrument. MODIS can take surface temperature measurements at a spatial resolution of 1km (the exact grid size is 0.928km by 0.928km), operating in the wavelength ranges of between $\sim$3.7-4.5$\mu$m and $\sim$10.9-12.3$\mu$m. In addition to surface temperature measurements that were used in this work, MODIS can take observations of cloud properties, water vapour, ozone, etc. Here MYD11A1 v006 (Wan et al., 2015) collection that provides daily LST measurements at a spatial resolution of 1km on a sinusoidal projection grid SR-ORG:6974 (takes a spherical projection but a WGS84 datum ellipsoid) is exercised. Daily global LST data is generated by first applying a split-window LST algorithm (Wan and Dozier, 1996) on all nominal (i.e. 1km at nadir) resolution swath (scene) with a nominal coverage of 5 minutes of MODIS scans along the track acquired in daytime, and secondly by mapping results onto integerized sinusoidal projection; for more details see Wan et al. (2015) and Figure 3. Validation of this product was carried out using temperature-based method over different land cover types (e.g. grasslands, croplands, shrublands, woody areas, etc.) in several regions around the globe (i.e. United States, Portugal, Namibia, and China) at different atmospheric and/or surface conditions; the best accuracy is achieved over United States sites with RMSE lower than 1.3K (Duan et al., 2019). At large view angles and in semi-arid regions the data product may have slightly higher errors due to rather uncertain classification-based surface emissivities and heavy dust aerosols effects. In addition, the MODIS cloud mask struggles to eliminate all cloud and/or heavy aerosols contaminated grid-cells from the clear-sky ones (LST errors in such grid-cells can be 4-11K and larger). Validation of this product over five bare ground sites in north Africa (in total 12 radiosonde-based datasets validated) showed that mean LST error was within $\pm$0.6K (with exception for one dataset, where mean LST error was 0.8K) and standard deviation of LST errors were less than 0.5K (Duan et al., 2019). In this work to reduce

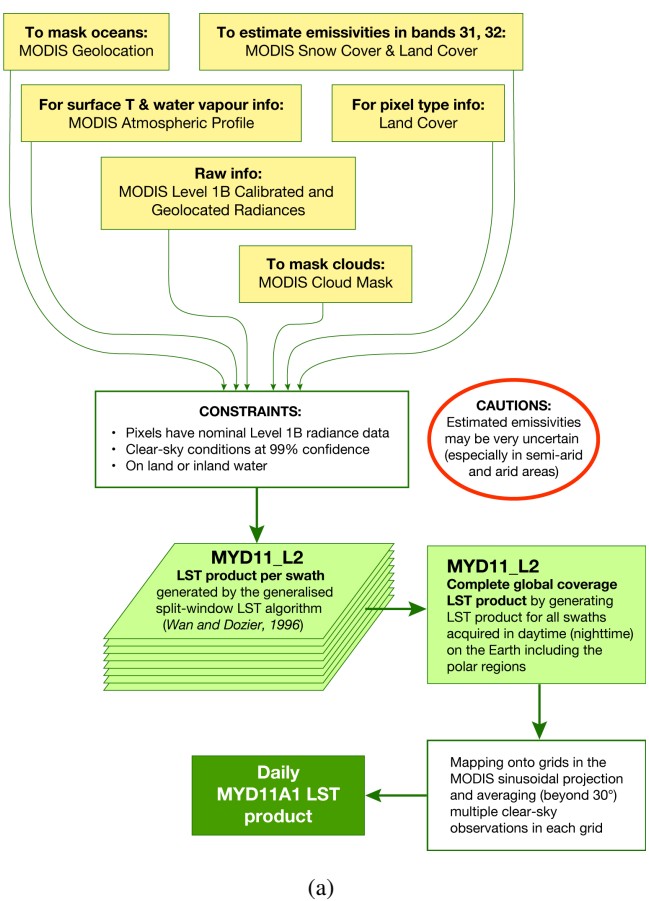

(a)

**Figure 3.** A brief step-by-step explanation of the LST algorithm for MYD11A1 v006 collection.

the amount of daily data over multiple years to store and manipulate, prior use LST data is (i) filtered to contain only cloud free data, and (ii) averaged to a 4km at the Equator resolution on a regular latitude-longitude grid, EPSG:4326 (note that only grid cells which have 8 or more valid observations at 1km resolution are averaged over, otherwise they are classified as missing data).

### 2.3 Joining the data

To join selected ERA5 global fields on a reduced Gaussian grid at $\sim$ 31km resolution (information in UTC, 24 hourly maps per day) with Aqua-MODIS global LST data on a regular latitude-longitude grid at 4km resolution (information in local solar time, 1 map per day), both datasets need to be at the same time space. First it is necessary to determine the absolute time (i.e. UTC) at which the MODIS observations were taken. Since in general all Aqua observations are taken at 1.30pm local solar time, it can be related to a UTC via observation longitude, following Eq. 1:

$$\text{UTC} = \text{Local solar time} - \frac{\text{longitude}}{15} \; , \tag{1}$$

where longitude is in degrees, and UTC is rounded to the nearest hour. This conversion is inexact since there is an additional correction as a function of the latitude, yet recommended by the official MODIS Products User's Guide (Wan et al., 2015); given the short orbital period of Aqua these additional higher order corrections are expected to be typically small and for our purposes can be neglected. Also, the assumption that all Aqua observations are taken at 1.30pm local solar time was checked (see Figure 4). The annually averaged mean time difference at 31km resolution (i.e. daily differences between local solar time of observations and 1.30pm at 1km resolution were first aggregated to 31km resolution using averaging, and then aggregated in time over a year) is 0.16 hours or 10 minutes, with mean absolute error (MAE) being 0.46 hours or 28 minutes and RMSE being 0.61 hours or 37 minutes (current values correspond 70N-70S region year 2019, but confirmed to be approximately identical for each year of 2016-2019 period). Since the temporal resolution of ERA5 data is hourly, the assumptions inherent to Eq 1 are sufficiently accurate. Over the poles (i.e. 90-70°N and 70-90°S) satellite sweeps overlap significantly and in general conversion becomes less accurate (daily time differences can reach more than $\pm$ 3.5 hours), so these areas were not included in the analysis.

Once Aqua-MODIS time of observation is converted to UTC, Aqua-MODIS data at $\sim$ 4km resolution is matched in time and space to ERA5 information in the following way:

1. Take a single Aqua-MODIS LST observation at a particular point on the MODIS grid;

2. Select ERA5 global hourly map matching Aqua-MODIS LST observation time in UTC;

3. Find the nearest point on the ERA5 grid to that MODIS grid point;

4. Repeat previous steps for every Aqua-MODIS observation;

5. Group matched data pairs by the ERA5 grid points, averaging over all the Aqua-MODIS observations that are associated with each ERA5 point.

At the end of this process selected ERA5 fields are mapped to a single Aqua-MODIS time of observation and Aqua-MODIS LST data is mapped (i.e. multiple Aqua-MODIS observations could be averaged over, see Figure 5a) to a reduced Gaussian grid at 31km resolution; averaged Aqua-MODIS observations are considered as ground truth (i.e. targets $y$) that VESPER is trying to predict. To better understand VESPER's grid-cell results at 31km resolution additional information was computed from Aqua-MODIS, namely (i) total number of valid observations per month and year (see Figure 5a), and (ii) average LST error based on Aqua-MODIA quality assessment (i.e. quality flag, see Figure 5b). Based on this additional information it can be concluded that areas with sparse number of observations in general have more uncertain LST values; exceptions are Alaska in United States and Anadyrsky District in Russia (area 30° east and west from 180°E around 70-60°N), deserts of Australia and Kalahari desert in Namibia, Botswana and South Africa, where majority of vast number of observations have only good or average quality.

For step (3) in the joining process, we use a GPU-accelerated k-nearest neighbours algorithm RAPIDS (v22.04.00), where "nearness" on the sphere between two points is measured via the Haversine metric, i.e. the geodesic distance $H$, following Eq.

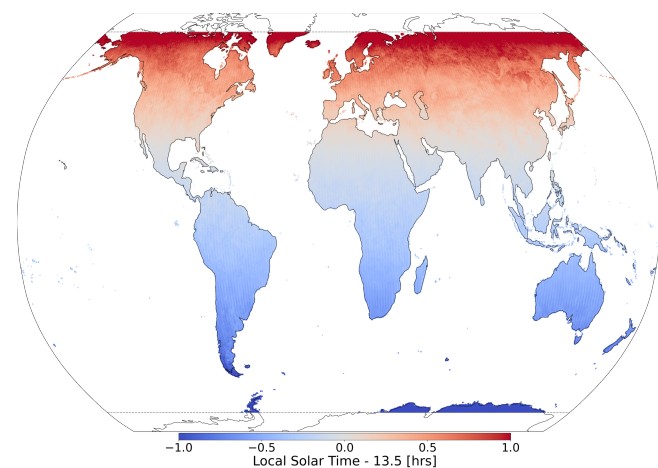

**Figure 4.** The annually averaged mean time difference of Aqua-MODIS and assumed local solar time of 1.30pm for the year 2019 at 31km resolution. Time differences are generally sub-hour and grow at greater latitudes, so data over 90-70°N and 70-90°S is excluded.

2:

$$H = 2\arcsin\left(\sqrt{\sin^2\left(\frac{\delta\theta}{2}\right) + \cos\theta_1\cos\theta_2\sin^2\left(\frac{\delta\phi}{2}\right)}\right) \tag{2}$$

for two points with coordinate latitudes $\theta_{1,2}$, longitudes $\phi_{1,2}$ and $\delta\theta = \theta_2 - \theta_1$ and $\delta\phi = \phi_2 - \phi_1$ .

## 2.4 Constructing a regression model

VESPER is trained to learn the mapping between features $x$ and targets $y$ (i.e. mapping ERA5 to MODIS), a regression problem. For this purpose a fully-connected neural network architecture (also known as a multi-layer perceptron), implemented in Tensorflow (Abadi et al., 2016) was used. Whilst more advanced architectures are available, for the purposes of this work the model is sufficient enough, which exhibits generally fast and dependable convergence. The networks built have differing number of nodes in the input layer, depending on the number of predictors (see Table 3). For all networks constructed we use 4 hidden layers and a layer width is half that of the input layer width. ADAM (Kingma and Ba, 2014) is used as an optimisation scheme, learning rate is set to $3 \times 10^{-4}$, and default values for the exponential decay rate for the 1st and 2nd moment estimates are set to 0.900 and 0.999 respectively. The network is not trained for a fixed number of epochs, but instead trained until the validation error reaches a minimum. Techniques for maximising the performance of a network via hyperparameter optimisation are now well established (Yu and Zhu, 2020; Bischl et al., 2021). However, for the purposes of this work no attempt to tune hyperparameters was made, just some reasonable default values were applied which were assumed to be "good enough". Some exploration of different hyperparameter configuration was undertaken, but for this data the prediction accuracy is mostly independent of the hyperparameter configuration, subject to standard and reasonable hyperparameter choices. Whilst a more advanced automatic hyperparameter optimization method may have enabled slightly higher performance of VESPER, our ultimate purpose is not to generate the most absolutely accurate prediction possible, but instead to have two predictive

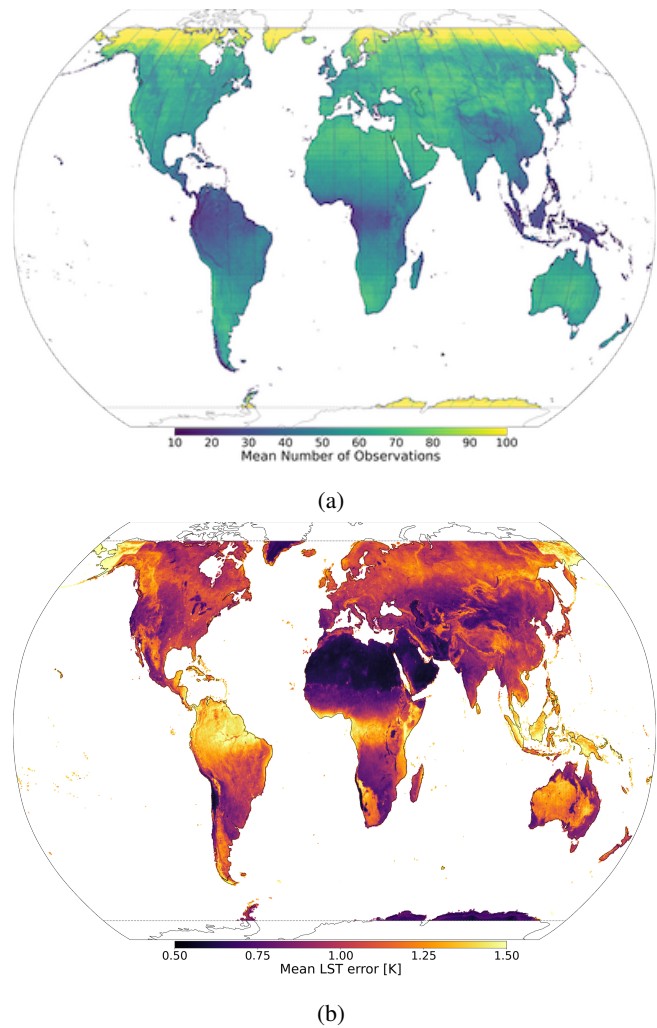

(a)

(b)

**Figure 5.** For 2019 at ~ 31km resolution: (a) Mean daily number of Aqua-MODIS observations mapped to each ERA5 data point. The swath of the Aqua satellite is clearly visible, with more observations over 70-60°N and 60-70°S areas as Aqua follows a polar orbit, south to north, and with less observations over Equator, complex orography areas (such as the Himalayas, the Andes and the Rocky Mountains), and the Siberian Tundra (due to increased cloud cover); (b) Average error in the Aqua-MODIS LST measurement. The raw Aqua-MODIS data at 1km resolution provides categorical LST errors with bins ≤ 1K, 1-2K, 2-3K and > 3K. When averaging to the coarser resolution a weighted average over the 1km grid-cells is computed, where the median bin value is used, and 5K for the > 3K bin. This information helps to understand that abundant number of observation does not automatically mean high quality of LST (e.g. Australia).

| Model | ERA5 atmospheric and surface fields | Main surface physio-graphic fields, V15 | Main surface physio-graphic fields, V20 | Additional surface physio-graphic fields |
|---|---|---|---|---|
| VESPER_V15 | ✓ | ✓ | - | - |
| VESPER_V15X | ✓ | ✓ | - | ✓ |
| VESPER_V20 | ✓ | ✓ | ✓ | - |
| VESPER_V20X | ✓ | ✓ | ✓ | ✓ |

**Table 3.** List of input files for different VESPER versions. c.f. Table 1

models which can be compared. In the result section below it will be shown that the variation in performance due to input feature modifications is far greater than the variation due to the hyperparameter choices.

VESPER was trained on selected atmospheric and surface model fields from ERA5 for 2016 (see Table 1), certain static version
of the surface physiographic fields (see Table 2), and Aqua-MODIS LST for 2016. Once VESPER was fully trained it was used to predict LST over the whole globe for 2019. Going forward, as a shorthand we will refer to VESPER trained using the e.g. $V15$ field set as VESPER_V15 (in general VM is a field set version and VESPER_VM is a VESPER model trained using the fields from the VM field set). See Table 3 for an explicit definition of all the VESPER models. The training and test years were chosen simply as recent, non-anomalous years so that the updated surface physiographic fields could be checked. All VESPER
versions are trained with ERA5 fields for 2016 and with main surface physiographic fields from V15 field set. Then depending on the version some or all additional surface physiographic fields (see Table 1) are added. VESPER's predictions can be compared to the initial ERA5 skin temperatures and actual Aqua-MODIS LST for 2019. Figure 6 shows the mean absolute errors (MAE) globally in the VESPER_V15 LST predictions, relative to the Aqua-MODIS LST along with the corresponding MAE in the predicted skin temperature from ERA5. We can see that VESPER_V15 was able to learn corrections to ERA5, especially
in the Himalayas and sub-Saharan Africa as well as Australia and the Amazon basin, leading to the globally averaged MAE reduction for predicted LST; the MAE relative to Aqua-MODIS LST, averaged over all grid points, was 3.9K for ERA5 and 3.0K for VESPER_V15.

As the focus of this study is lake-related fields, and lakes occupy only 1.8% of the Earth's surface and are distributed very
heterogeneously (Choulga et al., 2014), analysis of the results was restricted to areas where there have been significant changes in the surface lake physiographic fields. By "significant change" we mean a change in any of the surface field when going from V15 to V20 (and to V15X or V20X) of $\geq 10\%$ ($\geq 0.1$ for fractional fields); for example if lake or vegetation cover changed from 0.1 in V15 to 0.3 in V20 field set this change is classified as significant. The choice of $\geq 10$ % as a significance cut-off was adopted as it proved to be a good trade off between having a sufficient number of grid points to inspect and the strength
of the effect of changing the input field. As the cut-off % increases less points are selected, albeit with more severe changes to their surface fields, whereas when the cut-off % decreases more points are selected but it becomes more difficult to disentangle

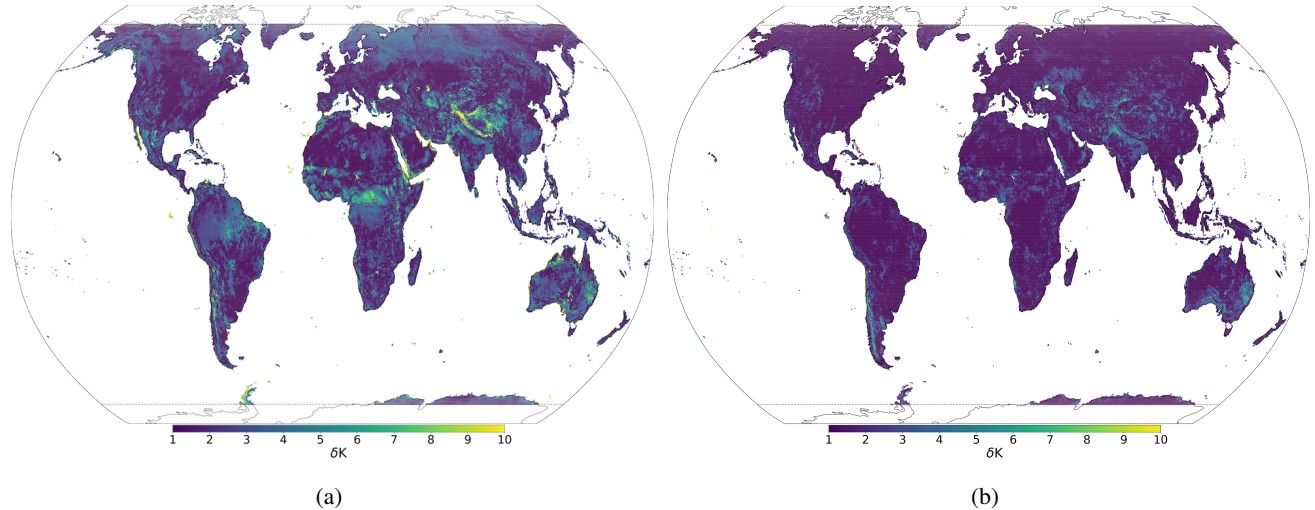

(a)                                                          (b)

**Figure 6.** Mean absolute error (MAE, $\delta K$) of LST predictions for 2019 at 31km resolution based on differences between (a) ERA5 skin temperature and Aqua-MODIS LST and (b) between VESPER_V15 (i.e. VESPER trained with V15 surface physiographic fields) and Aqua-MODIS LST. It can be seen that VESPER_V15 managed to learn corrections over regions with complex surface fields such as the Himalayas (lots of orography) sub-Saharan Africa (lots of vegetation) and the Amazon Basin (lots of water + vegetation).

the change in the prediction accuracy from VESPER's training noise (training noise is discussed below). Alternative cut-off % were briefly explored, but conclusions of the results remained broadly unchanged. All grid-cells selected for the analysis can be classified according to how the surface fields are updated when going from V15 to V20:

– **Lake Updates**. The change in the lake cover *cl* and lake depth *dl* are significant, but the changes in ocean and glacier *glm* fractions are not. This corresponds to grid-cells where lakes have been added or removed. Lake-Ground Updates is a sub-category where additional constraint that the change in the high/low vegetation fractions *cvh/cvl* are not significant is in place. This then corresponds to the exchange of lakes for bare ground, or vice versa.

   – **Vegetation Updates.** The change in the high vegetation fraction *cvh* is significant, but the change in lake cover *cl* is not
significant. This corresponds to grid-cells where large features like forests and woodlands have been updated, exchanged for bare ground or low vegetation.

   – **Glacier Updates**. The change in the glacier cover *glm* is significant. This corresponds to any areas where the fraction of glacier ice has been updated.

The training of a neural network is inherently stochastic - the same model trained twice with the same data can settle in
different local optima and so make different predictions. To make our conclusions robust against this training noise, each VESPER model is in turn trained 4 times. For each MODIS ground truth we then have 4 LST predictions per model. We define the training noise as the standard deviation, $\sigma$, in the VESPER predictions for the same input fields i.e. each VESPER_VM model will have a corresponding training noise $\sigma_{VM}$. To assess the changes of LST predictability due to the use of the updated

surface physiographic fields instead of V15 field set (default) we compare the mean absolute error (MAE) between different
VESPER models using the simple metric $\delta_{\mathrm{VM}}$:

$$\delta_{\mathrm{VM}} = \mathrm{MAE}_{\mathrm{VESPER\_VM}} - \mathrm{MAE}_{\mathrm{VESPER\_V15}} \tag{3}$$

where VM represents one of the field set versions V20, V20X or V15X, and MAE is computed over the whole prediction
period of 2019. In turn, the MAE is the error between the prediction of a VESPER model and the Aqua-MODIS LST, i.e.

$$\mathrm{MAE}_{\mathrm{VESPER\_VM}} = \frac{1}{N} \sum_{i=1}^{N} |\mathrm{LST}_{i,\mathrm{VESPER\_VM}} - \mathrm{LST}_{i,\mathrm{MODIS}}| \tag{4}$$

for total number of predictions $N$, within a given grid-cell classification. A negative $\delta_{\mathrm{VM}}$ then indicates that the VESPER_VM
LST prediction is more accurate than the VESPER_V15 prediction, and vice versa.

## 3  Results

### 3.1  Evaluation of updated lake fields

To understand if there is a way to automatically and rapidly assess the accuracy of updated and/or new surface physiography
fields, and if their use in the atmospheric model increase predictability, we can compare the prediction accuracy of different
VESPER_VM models. Generally VESPER's training noise is confirmed to be smaller than differences in LST predictions by
different VESPER configurations, so changes in LST predictability can be meaningfully attributed to the changes in surface
physiographic fields. Particular situations where the training noise becomes significant are discussed below.

As a first attempt lake-related information is assessed, namely lake cover (and land-sea mask and glacier cover as they are
used for lake cover generation) and lake mean depth, that were created from scratch using new up-to-date high-resolution input
datasets (see Table 2) for the V20 (and V20X) field set; other surface physiographic fields (see Table 1) were regenerated from
the same input sources as in the initial V15 field set, but taking into account that lake related fields were changed. In cases
when existing in V15 lake cover water was removed in V20, it could be replaced by any of high or low vegetation, glacier or
bare ground. We now analyse the results for each of the 4 categories of grid cell in detail (see Table 4 for the results of each
category aggregated over the whole globe).

#### 3.1.1  Category: Lake updates

The Lake Updates category shows significant improvements in LST predictability if using V20 field set instead of V15 –
prediction accuracy increased globally (over 1631 grid-cells) on average by 0.37K. For the lakes category, the training noise
in V20 was generally small $\sigma_{\mathrm{V20}} \sim 0.02$ K, with the V15 predictions a little more noisy with $\sigma_{\mathrm{V15}} \sim 0.07$ K, but this noise is
much less than the improvement - as can be seen in Fig. 7 every V20 iteration significantly outperforms every V15 iteration.
In Fig. 8 we plot the distribution of the mean LST error (averaged across each of the 4 trained VESPER iterations) for all lake

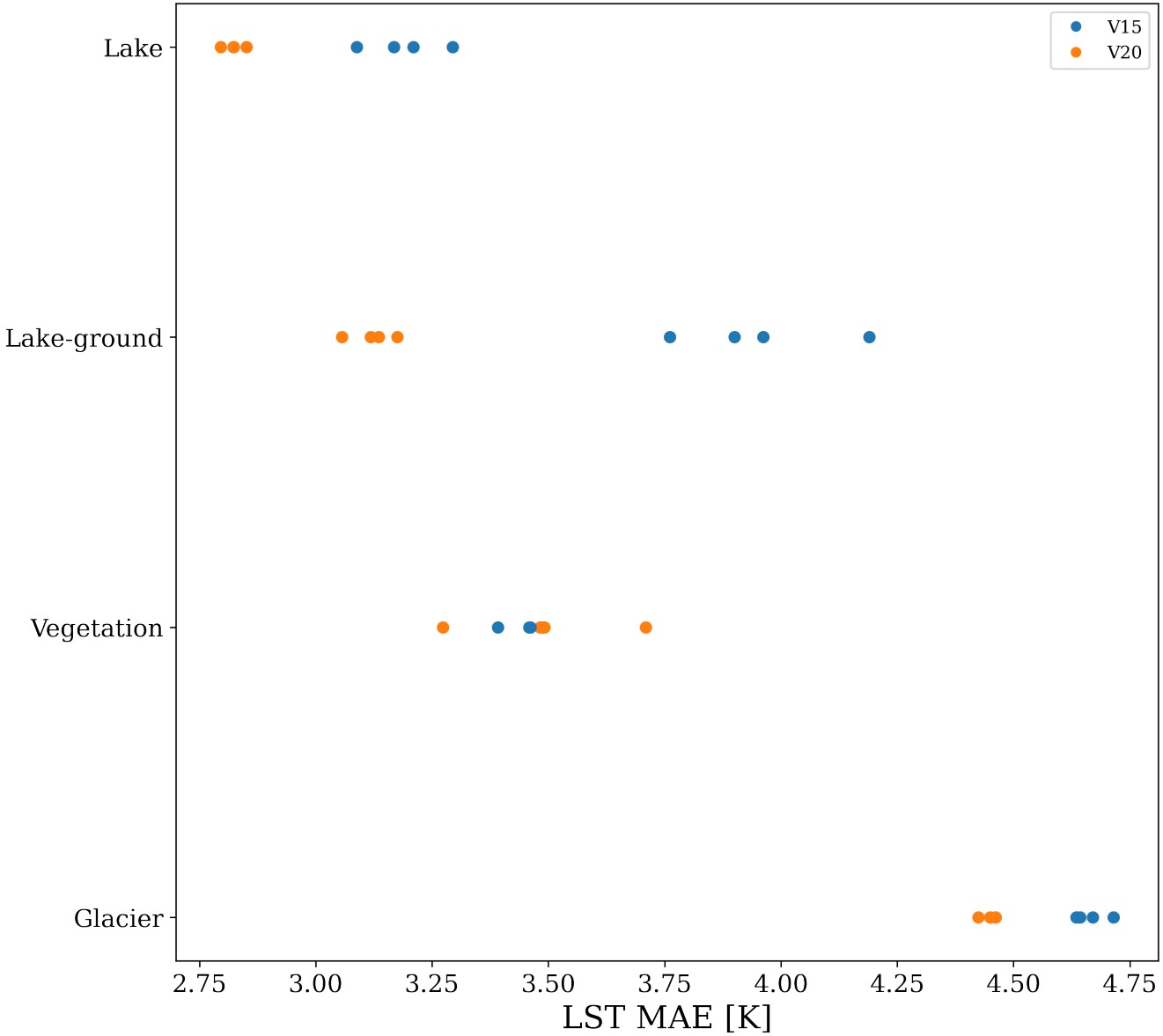

**Figure 7.** Distribution of prediction errors in the LST, for each of the 4 grid point categories, for each iteration of V15 and V20. For Lake, Lake-ground and Glacier categories the improvement in V20 relative to V15 is much greater than the intrinsic model noise, with all V20 predictions outperforming all V15 predictions. For the Vegetation category the predictions of V15 and V20 are much more noisy and it is difficult to draw any conclusions for the category as a whole.

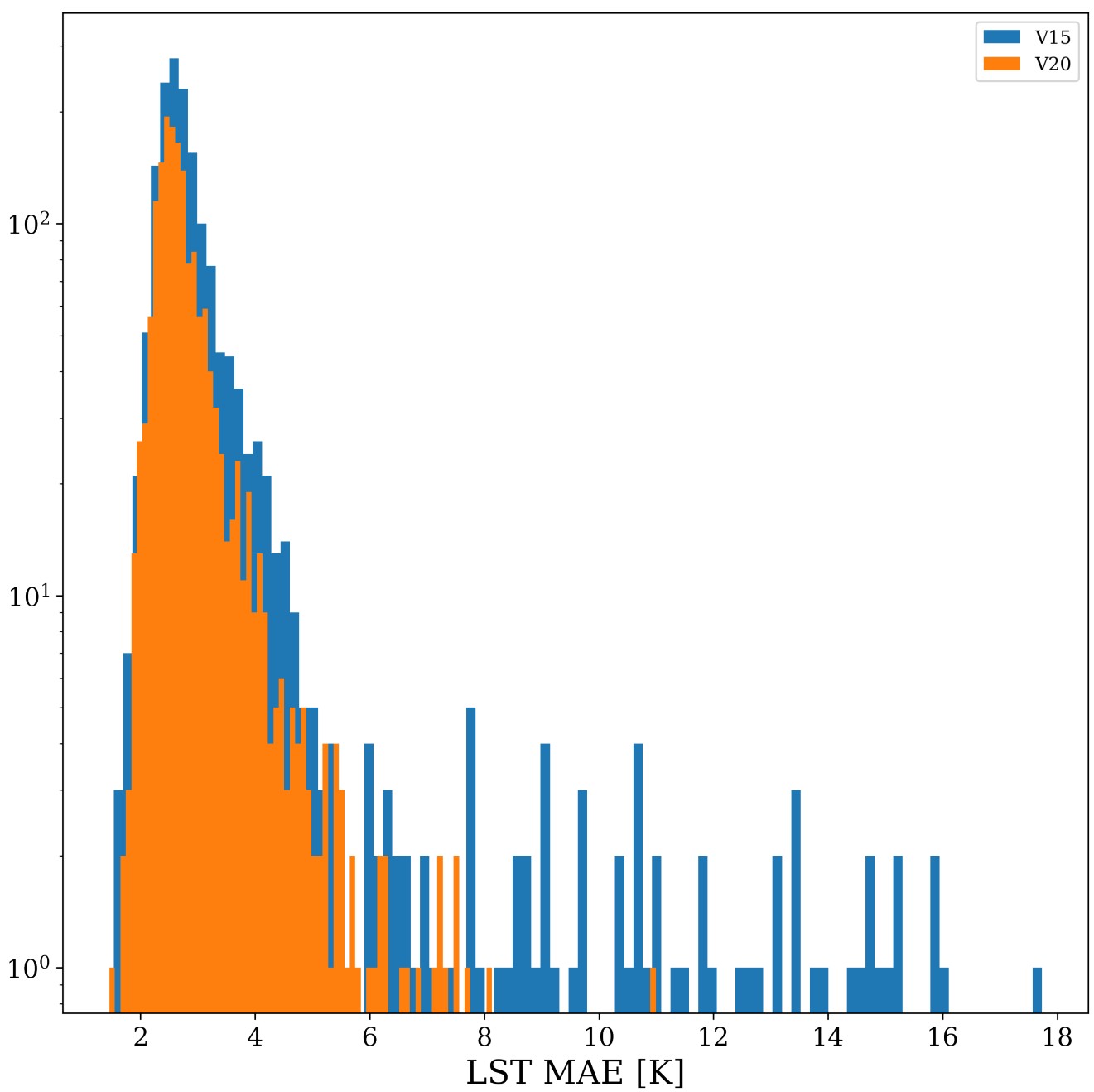

**Figure 8.** Distribution of prediction errors in the LST for all grid points in the Lake Updates category for VESPER_V15 and VESPER_V20. Each prediction errors is in turn the average of 4 trained iterations of the VESPER model. The predictions of VESPER_V20 are evidently and improvement over VESPER_V15, especially for grid points with large LST errors.

| Category | Number of grid cells | $\sigma_{\mathrm{VM}}, K$ | | | | $\delta_{\mathrm{VM}}, K$ | | |
| --- | --- | --- | --- | --- | --- | --- | --- | --- |
| | | V15 | V15X | V20 | V20X | V15X | V20 | V20X |
| Lake | 1631 | 0.07 | 0.02 | 0.02 | 0.02 | -0.20 | -0.37 | -0.37 |
| Lake-Ground | 546 | 0.15 | 0.05 | 0.04 | 0.06 | -0.56 | -0.83 | -0.84 |
| Vegetation | 58 | 0.04 | 0.10 | 0.15 | 0.21 | -0.00 | 0.04 | -0.00 |
| Glacier | 1057 | 0.03 | 0.08 | 0.02 | 0.06 | -0.01 | -0.22 | -0.28 |

**Table 4.** Globally averaged differences $\delta_{\mathrm{VM}}$ between mean absolute error (MAE) of VESPER_VM and VESPER_V15 LST for 2019 at 31km resolution (where M denotes V15X, V20, V20X field sets) per grid-cell category. Negative $\delta_{\mathrm{VM}}$ values indicate an increase of LST predictability due to the use of the updated surface physiographic fields instead of V15 field set (default), positive $\delta_{\mathrm{VM}}$ values indicate a decrease in the LST predictability and suggests the presence of erroneous information in the surface physiographic fields. Training noise values, $\sigma_{\mathrm{VM}}$, are generally much smaller than the variance between different VESPER configurations, indicating that changes in LST predictability are mainly due to changes in the surface physiographic fields. The quoted noise is the standard deviation of the prediction errors of Fig. 7.

grid points, for both V15 and V20. Evidently the V20 field significantly improve the high tail behaviour relative to V15, as well as shifting the median of the distribution to lower errors. Particular regions where the V20 physiographic fields notably improved performance were in Australia and the Aral sea (e.g. Fig. 9). These are two major regions where ephemeral lakes were removed and inland water distribution made up-to-date, as discussed in Section 2.2.1. In addition to the areas with a notable improvement in the prediction accuracy, there are some noteworthy regions where the predictions got worse (see red points in Figure 9) suggesting inaccuracies or lack of information in the updated surface physiographic fields. A few of the most noteworthy grid-cells (see red points highlighted with green circles in Figure 9 and also Figure 11) are:

– **Northern India.** This grid-cell lies in the state of Gujarat, India, close to the border with Pakistan. Here $\delta_{\mathrm{V20}} = +4.21$, with $\sigma_{\mathrm{V15}} = 2.54$ and $\sigma_{\mathrm{V20}} = 0.416$. The lake fraction was increased from 0.59 in V15 to 0.71 in V20 field set, along with the lake depth increase from 2.58m to 3.76m. However, this point lies on a river delta within the Great Raan of Kutch, a large area of salt marshes (see Figure 10a), known for having highly seasonal rainfall, with frequent flooding during the monsoon season and a long dry season. The surface itself also undulates with areas of higher sandy ground known as medaks, with greater levels of vegetation. It is evidently a complex and highly time variable area and additional static fraction of fresh water provided via V20 field set is not sufficient.

– **Salt Lake City, North America.** This grid-cell lies within the Great Salt Lake Desert, just to the west of the Great Salt Lake, Utah, US. Predictions of VESPER_V20 are worse than VESPER_V15, with $\delta_{\mathrm{V20}} = +2.91$ ($\sigma_{\mathrm{V15}} = 0.26$ and $\sigma_{\mathrm{V20}} = 0.92$). Whilst the training noise is significant here, it is less than the $\delta_{\mathrm{V20}}$ value, and we can see from Fig 11 that the VESPER_V20 predictions consistently underperform the VESPER_V15 predictions. The lake fraction was completely removed from over 0.50 in V15 to 0.00 in V20 field set, meaning that the grid-cell is fully covered with bare ground in V20 field set. Whilst this area primarily is bare ground, satellite imagery also suggests the presence

of a presumably highly saline lake (see Figure 10b); in addition area has a large degree of orography and high eleva-

tion ($\sim$1300m) which probably further complicates the surface temperature response. A more accurate description that

accounts for the seasonality of the surface water and the salinity is necessary here.

– **Tanzania**. There are two grid-cells of interest at the centre and northern edge of Lake Natron, which itself lies to south-

east of Lake Victoria, in Tanzania. For both these points VESPER_V20 predictions are less accurate than VESPER_V15;

for the central point ($\delta_{V20} = +2.45, \sigma_{V15} = 0.12$ and $\sigma_{V20} = 0.81$, see also Figure 10c) the lake fraction was increased

from 0.04 in V15 to 0.39 in V20 field set; for the northern edge point ($\delta_{V20} = +1.57, \sigma_{V15} = 0.13$ and $\sigma_{V20} = 0.51$)

the lake fraction was also increased in V20 comparing to V15 field set along with a small decrease ($\sim$0.1) in the low

vegetation fraction. However, Lake Natron is a highly saline lake that often dries out, with high temperatures, high levels

of evaporation and irregular rainfall. It is a highly complex and variable regime that is not well described by simply

increasing the fraction of permanent fresh water, and indeed results suggest that with current lake parametrization scheme

it may be beneficial to keep the lake fraction low or introduce extra descriptor, e.g. salinity.

– **Algeria**. This grid point lies in Algeria, at the northern edge of the Chott Felrhir, an endorheic salt lake ($\delta_{V20} = +2.20$,

$\sigma_{V15} = 0.41$ and $\sigma_{V20} = 0.49$). Similar to the Great Salt Lake Desert, the lake fraction was completely removed from

0.33 in V15 to 0.0 in V20. However, Chott Felrhir goes through frequent periods of flooding where the lake is filled by

multiple large wadi, and corresponding dry periods where the lake becomes a salt pan. As with the Great Salt Lake Desert

it is also a highly variable, complex area that may require additional consideration of the salinity and the seasonality.

– **Lake Chad** This grid point contains Lake Chad, a freshwater endorheic lake in the central part of the Sahel ($\delta_{V20} = +1.74$, $\sigma_{V15} = 0.33$ and $\sigma_{V20} = 0.98$). Here the lake fraction was modestly reduced from 0.63 to 0.47. However, Lake

Chad is again a highly time variable regime with seasonal droughts and wet seasons. It is a marshy wetland area but

the vegetation fractions in both V15 and V20 here are zero. Satellite imagery also shows a large fraction of the surface

covered by water and vegetation (Figure 10e).

– **Al Fashaga** This grid point lies in a disputed region between Sudan and Ethiopia called Al Fashaga, close to a tributary

of the Nile ($\delta_{V20} = +0.94$, $\sigma_{V15} = 0.14$ and $\sigma_{V20} = 0.29$). The updated V20 fields increased the lake fraction at this

point from 0 to 0.14. The grid cell contains the Upper Atbara and Setit Dam Complex. However, the dam was only

recently completed in 2018 - during the training period the damn was still under construction. Consequently whilst the

V20 field may be more accurate at the current time, during the period the model was training the V15 field was more

accurate, since the damn was not yet built.

– **Lake Tuz**. This grid cell contains a large fraction of Lake Tuz as well as the smaller Lake Tersakan, saline lakes in

central Turkey ($\delta_{V20} = +0.85, \sigma_{V15} = 0.25$ and $\sigma_{V20} = 0.34$). Here the updated physiographic field effectively removed

all lake water, with the lake fraction decreasing from 0.14 to 0.005. Whilst the lake is shallow and does dry out in the

summer, there is also a large fraction of surface water present (e.g. Fig 10d) and it is an over correction to completely

remove all lake water at this point.

– **Lake Urmia**. This grid cell contains Lake Urmia, which is another saline lake in Iran ($\delta_{V20} = +0.81$, $\sigma_{V15} = 0.12$ and $\sigma_{V20} = 0.73$). The updated physiographic fields decreased the lake fraction at this point from 0.77 to 0.39. This was in response to the shrinking of Lake Urmia due to long-timescale droughts and the damming of rivers in Iran. However, this drought broke in 2019 and Lake Urmia is now increasing in size again - satellite imagery now shows a large fraction of the grid cell covered by water (Figure 10f).

The Lake-Ground Updates sub-category, which restricts analysis to only points with no significant change in the vegetation, allows us to more clearly see the effect of adding/removing water on/from bare ground. This sub-category shows even larger improvements in LST predictability if using V20 field set instead of V15 (see Table 4) – prediction accuracy increased globally (over 546 grid-cells) on average by 0.83K ($\sigma_{V15} = 0.15$ and $\sigma_{V20} = 0.04$, see also Figure 7). This indicates that whilst the updated lake fields are globally accurate and informative, providing on average over the globe, over a year, nearly an extra Kelvin of predictive performance, the updates to the vegetation fields tamper this performance gain, indicating potential problem with the vegetation fields

### 3.1.2 Category: Vegetation Updates

The Vegetation updates category, restricts analysis to grid points with significant change to high vegetation cover, where the high vegetation cover is substituted with either low vegetation or bare ground, and vice versa. For this category the prediction accuracy of V20 decreased globally (over 58 grid-cells only) on average by 0.04K. However, this shift is much smaller than the training noise between successive VESPER iterations ($\sigma_{V15} = 0.04$, $\sigma_{V20} = 0.15$) and so it is hard to make definitive statements about the performance of the updated vegetation physiographic fields as a whole (see e.g. Fig 13). The best we can say is that the updated V20 vegetation fields offer no global improvement in the LST prediction accuracy.

If we isolate our analysis to individual grid points where the training noise is small (highlighted by $*$ points in Fig 13) we can discern that there are multiple locations where the high vegetation fraction was decreased (often quite drastically to zero), specifying that there should just be bare ground, but thorough inspection of these areas with satellite imagery revealed that they should in fact be covered with high vegetation (see e.g. Figure 12) and that updating the V20 high vegetation cover was erroneous for these grid-cells. Moreover, for this subset of less noisy grid points, the strength of the drop in LST predictability in VESPER_V20 comparing to VESPER_V15 is approximately linearly dependent to the degree of reduction in high vegetation fraction, when the vegetation is replaced with bare ground (i.e. $\delta_{V20}$ is maximally positive when the grid-cell that was fully covered with forest becomes fully covered with bare ground – high vegetation cover is reduced to zero). These erroneous grid-cells in V20 vegetation fields are likely to appear during the interpolation. The errors in these regions will in turn corrupt the LST predictions and mitigate the gain from a more accurate representation of the lake water. The majority of grid cells in this category (57/58) are modified in this way where the high vegetation fraction is severely reduced, however due to the large degree of training noise and the small number of points, it is difficult to draw any definitive conclusions for the category as a whole.

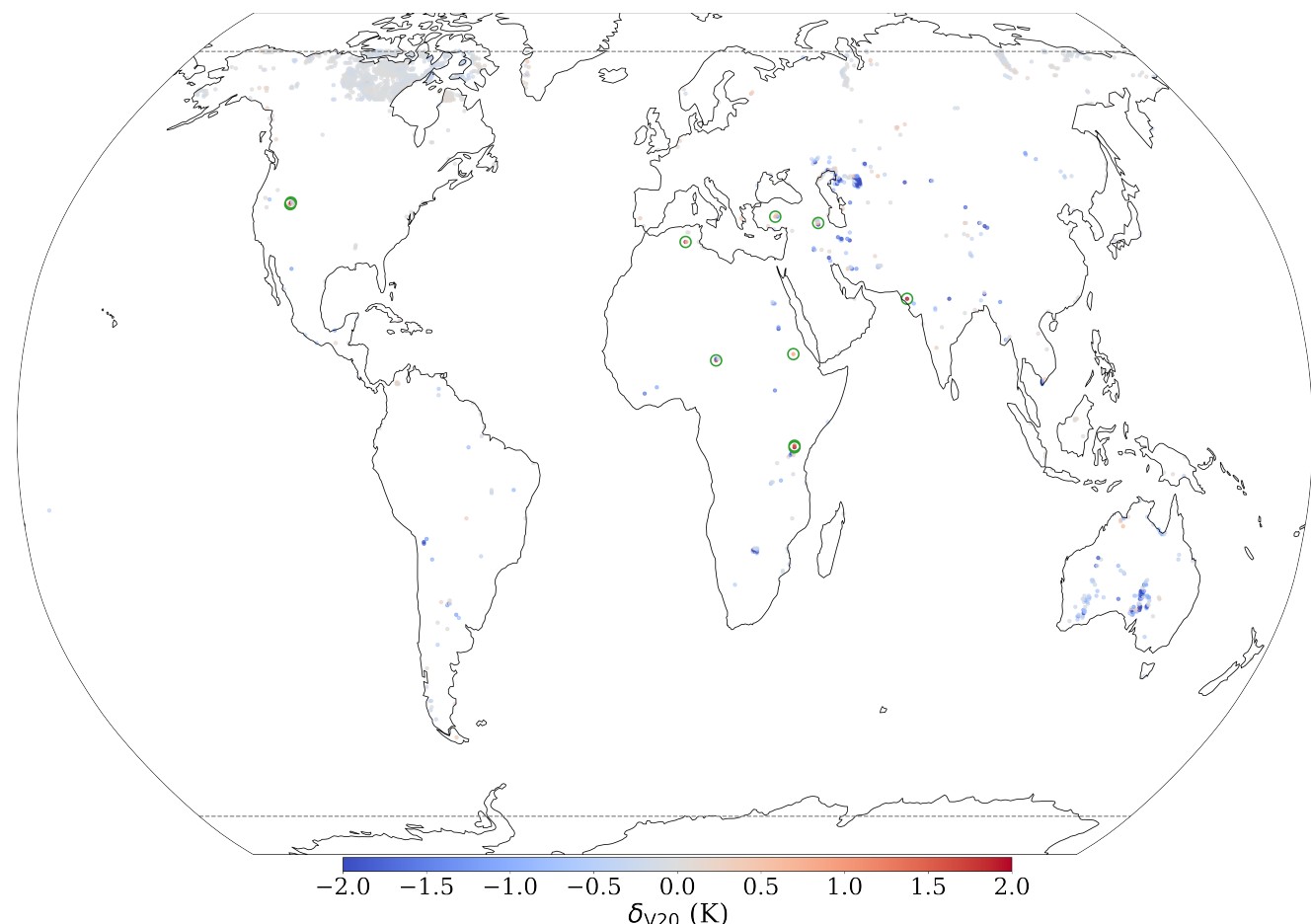

**Figure 9.** Differences in the prediction error MAE, between VESPER_V20 and VESPER_V15, (i.e. $\delta_{V20}$),for 2019 at 31km resolution for 'Lake Updates' category (i.e. where lake cover changed significantly). Generally, VESPER_V20 LST predictions are more accurate, for example in the Aral sea and Australia, indicating that V20 field set is informative and accurate. Particular points where VESPER_V20 LST prediction gets notably worse compared to VESPER_V15 are highlighted with green circles and discussed in the text.

### 3.1.3 Category: Glacier Updates

The Glacier Updates category in general shows improvement in LST predictability in VESPER_V20 comparing to VES-PER_V15 (see Table 4) – prediction accuracy increases globally (over 1057 grid-cells) on average by 0.22K ($\sigma_{V15} = 0.03$, $\sigma_{V20} = 0.02$), most notably around the Himalayas, the land either side of the Davis strait, as well as British Columbia and the Alaskan Gulf. Analogous to the Lakes Updates category whilst the introduction of the V20 glacier cover generally improves LST predictions, there is a small selection of grid points where the prediction gets worse. These are heavily concentrated in

the southern hemisphere, in particular on the south-western edge of South America and the South Shetland Islands (which lie closer to Antarctica), and some parts of the Himalayas. This deterioration in performance in these areas is not due to erroneous

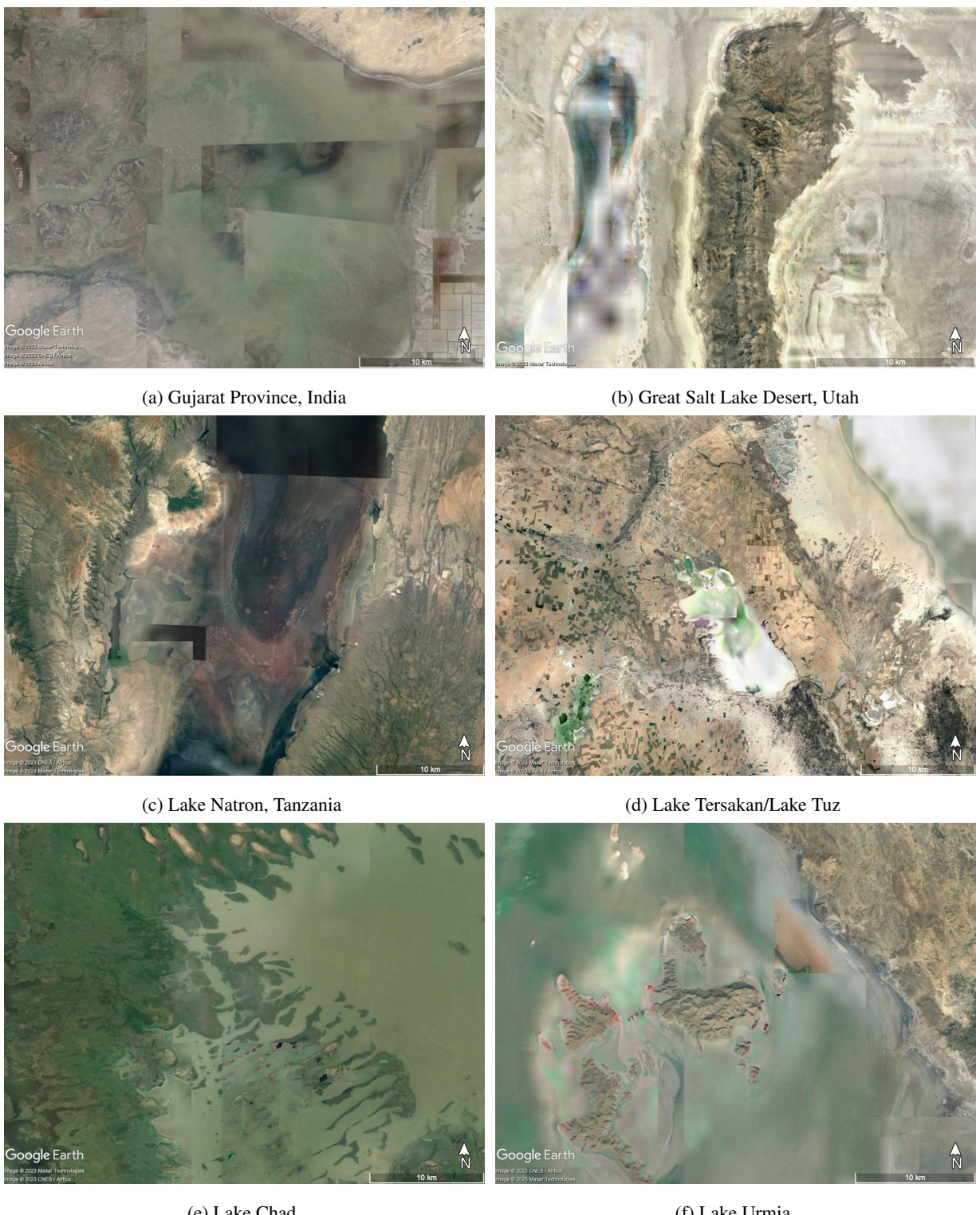

(a) Gujarat Province, India

(b) Great Salt Lake Desert, Utah

(c) Lake Natron, Tanzania

(d) Lake Tersakan/Lake Tuz

(e) Lake Chad

(f) Lake Urmia

**Figure 10.** A selection of satellite imagery of some of the problematic Lake Updates points highlighted in Fig. 9 where the V20 predictions are worse than the V15 predictions. Generally the updated V20 fields remove water, only considering permanent water. However these regions have highly time variable waters, which are better captured on average by the V15 fields. The images are centred on the grid box coordinates.

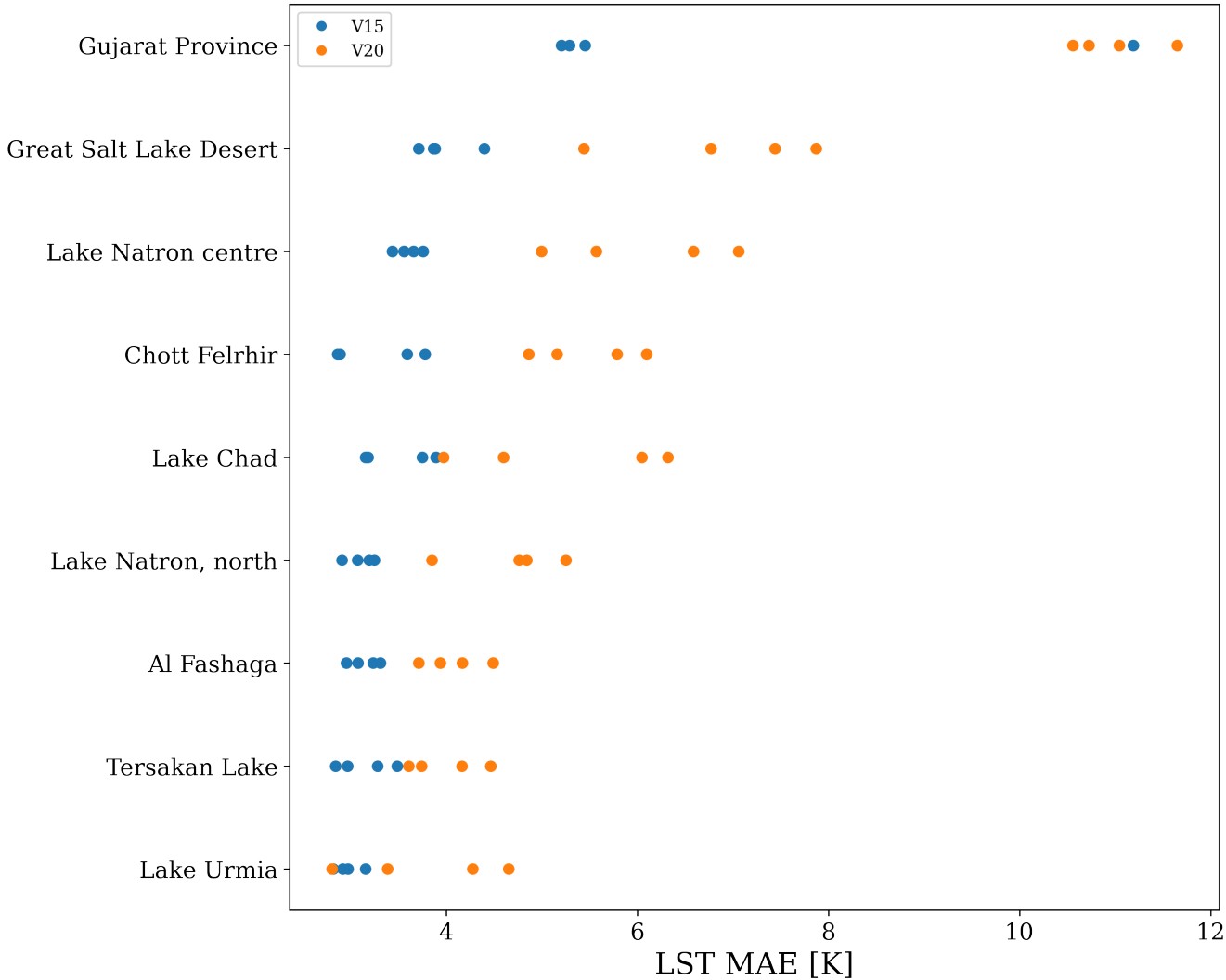

**Figure 11.** As Fig. 7 for selected locations in the lakes grid point category where the added V20 data results in worse predictions when compared to V15.

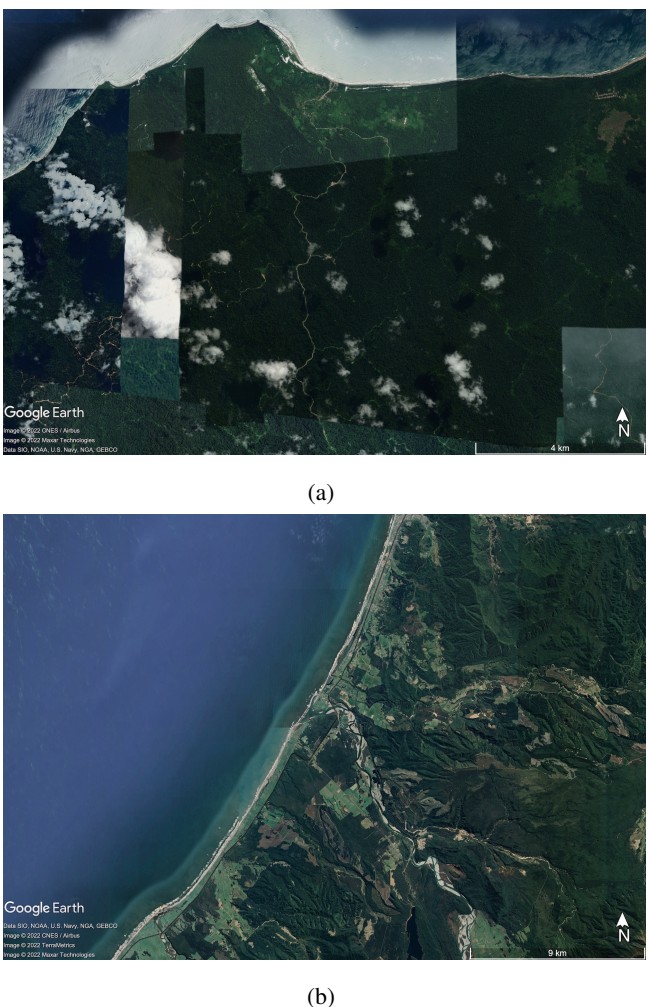

(a)

(b)

**Figure 12.** Satellite imagery of grid-cells in (a) Siberut Island, Indonesia and (b) South Island, New Zealand. For both grid-cells according to the updated V20 field set there should be no vegetation, just bare ground. VESPER identified these erroneously updated areas.

update of V20 glacier cover, but related to the Aqua-MODIS data (i.e. sparse availability due to clouds, and less certain due to orography, see Figure 5a). Consequently, VESPER finds it difficult to make accurate predictions in this region and for these points there is often a large degree of training noise, with considerable overlap between VESPER_V15 and VESPER_V20. If

grid- cells with scarce amount of Aqua-MODIS observations (i.e. mean number of Aqua-MODIS observations per day over the year per ERA5 grid-cell is >50) are removed from the analysis then the worst performing grid-cells become excluded, yet a few areas where VESPER_V20 underperforms VESPER_V15 remain. For example, there is a grid-cell in Chilean Patagonia that contains the Calluqueo Glacier, close to Monte San Lorenzo where $\delta_{V20} = 2.49$ ($\sigma_{V15} = 0.38$, $\sigma_{V20} = 0.62$). This grid-cell has been updated in V20 field set comparing to V15 by strongly increasing glacier cover from 0.0 to 0.44), decreasing low

vegetation cover (from 0.22 to 0.12) and high vegetation cover (from 0.16 to 0.09) as well as modestly decreasing lake cover

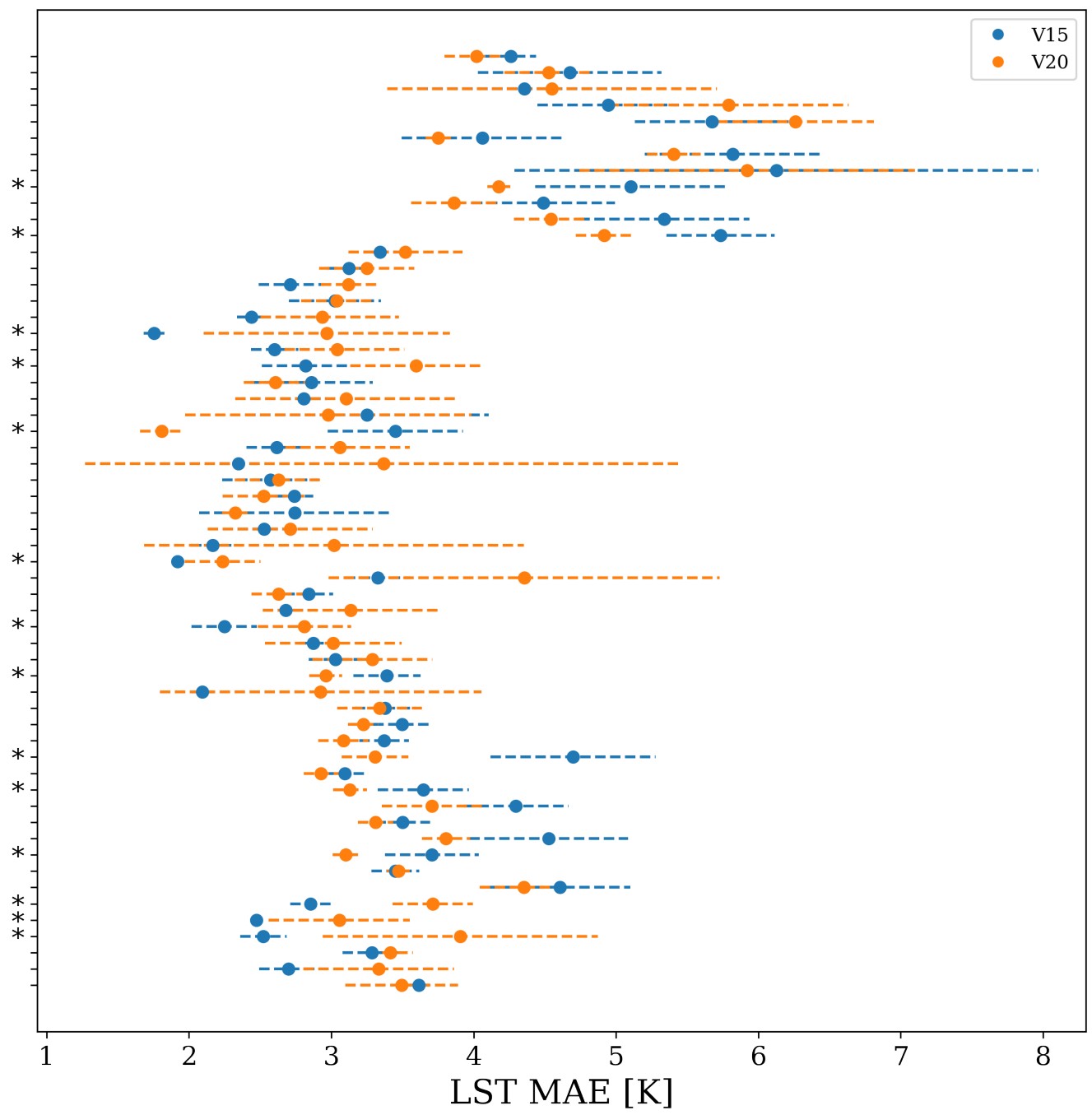

**Figure 13.** Distribution of prediction errors in the LST, for VESPER_V15 and VESPER_V20, for all 58 grid points in the vegetation category. There is evidently a large degree of noise, with predictions from both generations of VESPER model highly overlapping. Points with reduced training noise are highlighted with a ∗.

(from 0.02 to 0.007). According to satellite imagery (see Figure 14a) the glacier only occupies a small fraction of the overall grid-cell, and the updated glacier cover may have been an over correction. Moreover, this is an complex orographic area with snowy mountain peaks at high altitude and deep valleys, therefore the temperature response due to the glacier feature could be atypical compared to e.g. the Alaskan Gulf or the Davis straight. There is also substantial vegetation cover in the valleys that may not be being properly described. A similar point is in the Chilean Andes (see Figure 14b), by the Juncal Glacier with $\delta_{V20} = 1.26$ ($\sigma_{V15} = 0.68$, $\sigma_{V20} = 0.29$). Here V20 glacier cover was increased to 0.25 compared to 0.00 in V15. Again, this is may have been an over correction, as the glacier constitutes only a small fraction of the grid cell. As with the Calluqueo Glacier this is also an area with lots of orography and so could have an atypical temperature response. For both of these points VESPER managed to identify potential inaccuracies in updated glacier cover, and once again proved itself as a useful tool for quality control of surface physiographic fields.

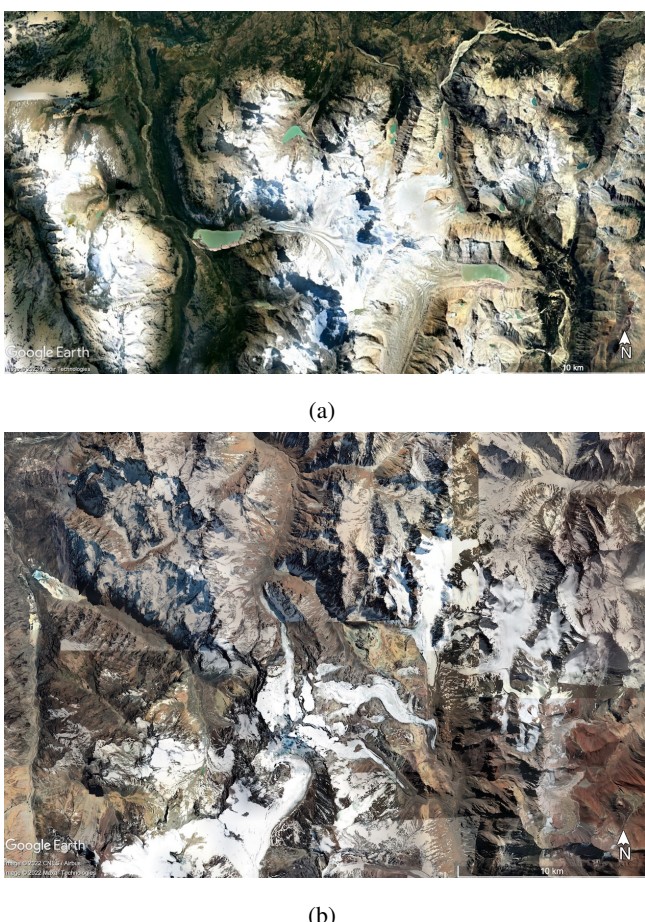

(a)

(b)

**Figure 14.** Satellite imagery of (a) Calluqueo Glacier, Patagonia, and (b) Juncal Glacier, Chile. In the updated V20 field set, the assumption for region (a) is almost half ice cover with little vegetation, for region (b) is quarter covered with glacier; these assumptions seem to be insufficiently accurate or informative, as identified by VESPER.

## 3.2 Evaluation of new lake fields: Monthly water & salt lakes

From the examples above it is evident that VESPER enables the user to quickly identify regions where the update to surface physiographic fields was beneficial (e.g. Aral Sea) and where it was not (e.g. Lake Natron). In turn, areas where LST predictions do not improve as expected can be inspected and erroneous or sub-optimal representations of the surface physiographic fields identified. This then provides key information on how and where to introduce additional corrections to better represent these more challenging or complex regions. Some of these problematic areas are now explored in more details and additional surface physiographic fields introduced with help of VESPER.

Particular regions where VESPER was struggling to make accurate LST predictions – especially with the updated V20 field set which only include permanent water – were either areas with a large degree of temporal variability (e.g. lakes which flood and dry out periodically) or else areas with saline rather than freshwater lakes. Clearly if the size, shape and depth of a lake are changing over the course of the year, these are going to be hugely significant factors in modelling the lake temperature response. Similarly, saline lakes behave very differently to freshwater lakes since increased salt concentrations affect the density, specific heat capacity, thermal conductivity, and turbidity, as well as evaporation rates, ice formation and ultimately the surface temperature. These two properties of time variability and salinity are often related; it is common for saline lakes to flood and dry out over the course of the season, which naturally also affects the relative saline concentration of the lake itself.

Currently, neither VESPER_V15 or VESPER_V20 have any information regarding the salinity of the lakes or their time variability. Indeed, FLake is specifically a fresh water lake model! This information can be introduced by including a global saline lake cover and monthly varying lake cover as additional VESPER's input features, and then using VESPER to rapidly assess the accuracy of these new surface physiography fields and evaluate if their use in the model increase LST predictability. We define an additional models (see Table 3 for a summary of all VESPER models used in this work); VESPER_V20X uses the same field set is the same as VESPER_V20 but with additional saline lake cover and monthly-varying lake cover. The results of this model in comparison with VESPER_V15 and VESPER_V20 is summarised in Tables 4, 5 . We will now explore the influence of the additional saline maps and monthly lake maps in more detail.

### 3.2.1 Category: Lake updates

The Lake Updates category shows no significant difference in LST predictability globally when using the V20X field set instead of V20, with $\delta_{V20X} = \delta_{V20} = -0.37$ (comparable training noise). For the Lake-ground category, there is a modest increase, with $\delta_{V20X} = -0.84$ compared to $\delta_{V20} = -0.83$ but this is within the training noise. For some of the problematic lake grid-cells highlighted in Table 5, the addition of saline maps and monthly lake maps does improve the LST predictability relative to VESPER_V20. For the Great Salt Lake Desert, Chott Fehlrir, Lake Chad and Lake Urmia, VESPER_V20X is a notable improvement over VESPER_V20, with $\delta_{V20X} = 0.248, 0.726, 0.029, 0.22$ respectively. The difference in $\delta_{V20X}$ and $\delta_{V20}$ for these points is greater than the training noise. If we take as a case example the grid point in the Great Salt Lake Desert,

| Category | Grid-cells/location | $\sigma_{\mathrm{VM}}, K$ | | | | $\delta_{\mathrm{VM}}, K$ | | |
|---|---|---|---|---|---|---|---|---|
| | | V15 | V15X | V20 | V20X | V15X | V20 | V20X |
| Lake | Gujarat Province, India | 2.54 | 1.12 | 0.42 | 1.04 | -1.26 | 4.21 | 5.24 |
| | Great Salt Lake Desert, Utah | 0.26 | 0.41 | 0.92 | 0.62 | -0.18 | 2.92 | 0.25 |
| | Lake Natron centre, Tanzania | 0.12 | 1.48 | 0.81 | 0.53 | 1.35 | 2.45 | 2.61 |
| | Lake Natron north, Tanzania | 0.13 | 0.37 | 0.51 | 0.18 | 0.72 | 1.57 | 1.24 |
| | Chott Felrhir | 0.41 | 0.57 | 0.49 | 0.58 | 0.34 | 2.20 | 0.73 |
| | Lake Chad | 0.33 | 1.21 | 0.98 | 0.96 | 0.29 | 1.74 | 0.03 |
| | Al Fashaga | 0.14 | 0.08 | 0.29 | 0.42 | -0.24 | 0.94 | 1.06 |
| | Tersakan Lake | 0.25 | 0.20 | 0.34 | 0.38 | -0.00 | 0.85 | 0.99 |
| | Lake Urmia | 0.12 | 0.54 | 0.73 | 0.32 | 0.54 | 0.82 | 0.22 |
| Glacier | Calluqueo Glacier, Patagonia | 0.38 | 0.62 | 1.60 | 0.73 | 0.08 | 2.49 | 0.32 |
| | Juncal Glacier, Chilean Andes | 0.68 | 0.29 | 1.06 | 0.36 | 0.11 | 1.26 | 1.20 |

**Table 5.** As Table 4 for specific grid points discussed in the text where the VESPER_V20 predictions are worse than VESPER_V15 (i.e. $\delta_{\mathrm{V20}}$ is positive).

the improvement in using VESPER_V20X over VESPER_V20 is 2.667K $\pm 1.10$ K. At this point there is a strong correction from the monthly lake maps (mean value 0.16) and the salt maps (mean value 0.56). This improvement is to be expected given the known strong salinity and time variability in the region, and so it is a nice confirmation to have these updated fields verified by VESPER. It is also notable that the variation in the monthly lake maps at this point is very large, with a standard deviation in the lake fraction over 12 months of 0.18. At the start of the year the corrections from the monthly maps are very large, then as the year progresses the magnitude of the corrections generally decreases as the lake dries out. Such a large variation is again difficult to ever capture with a static field.

It is however notable that a) for all of the problematic lake points that we have discussed $\delta_{\mathrm{V20X}}$ is positive and b) there are multiple points (e.g. Gujarat province) where VESPER_20X exhibits no improvement over VESPER_V20 within training noise. Given all the extra information provided to the more advanced VESPER_20X model this is unusual; it suggests that either i) some of the additional information is erroneous in these regions, or else ii) the temperature response is atypical to the rest of the globe. For point ii), this means that the additional information is not predictive in these regions. Including this additional information in our neural network increases the complexity of the model which may in turn increase its training noise. This is likely the reason behind point b) - the updated fields are not sufficiently informative but do increase the training noise and so we see no improvement from using VESPER_V20X. For example, for Gujarat province $\sigma_{\mathrm{V20}} = 0.416$, but $\sigma_{\mathrm{V20X}} = 1.04$. In order to explore the hypothesis of point i) we train one further model, VESPER_V15X (again, see Table 3 for a summary of all VESPER models used in this work). This VESPER iteration is analogous to VESPER_V20X, being simply the VESPER_V15 model with the additional monthly maps and salt lake fields included. Importantly it does not have the updated physiographic

correction fields from V20. Globally, this model performs worse that the V20 models, as we might expect - for example in the Lake Updates category $\delta_{\text{V15X}}$ = -0.20 ($\sigma_{\text{V15X}} = 0.02$) compared to $\delta_{\text{V20}}$= -0.37 K. However, VESPER_V15X does perform well at a number of the these problematic lake points (see Table 5). For 7 out of the 9 selected lake points, VESPER_V15X outperforms VESPER_V20X. For example in Gujarat province the improvement in using V15X over V20X is $6.5K \pm 1.53$. This suggests that our hypothesis for point i) is correct and that for some grid points the V20 fields are less accurate than the V15 fields. For a subset of points VESPER_V15X also outperforms VESPER_V15 (e.g. for Gujarat province $\delta_{\text{V15X}} = -1.26$) but the difference is typically within or close to the training noise (e.g. for Gujarat $\sigma_{\text{V15X}} = 1.12$) and so it is hard to draw any strong conclusions. These examples illustrates again how VESPER can identify particular regions where the fields are inaccurate, as well as emphasising the need more generally for accurate descriptions of seasonally varying inland water and saline lake maps in Earth system modelling.

### 3.2.2 Category: Vegetation Updates

Whilst the Vegetation Updates category explicitly deals with areas where the lake fraction does not change when going from V15 to V20, many of the grid points in this category do contain some kind of waterbody, often lying close to the coast or else containing lakes or large rivers. Information on the salinity and temporal variability of these water bodies could then influence the prediction accuracy. By providing the additional information in VESPER_20X, the error relative to VESPER_V15 is reduced modestly to $-3 \times 10^{-4}$ although as we saw before with the vegetation category the noise is large $\sigma_{\text{V20X}} = 0.21$ and so it is difficult to draw any further definitive conclusions. Similar arguments apply to VESPER_15X.

### 3.2.3 Category: Glacier Updates

We would expect the additional information provided by the V20X fields to be particularly effective for glacial grid points. Glacier ice is naturally found next to waterbodies which freeze and thaw over the year, and the salinity of water will also influence this freezing. Therefore accurate additional information from the monthly lake maps and the saline maps should prove useful in these more time variable regions. We do observe a small improvement globally, with $\delta_{\text{V20X}} = -0.28$ compared to $\delta_{\text{V20}} = -0.22$, however this difference is comparable to the training noise $\sigma_{\text{V20X}} = 0.06$. This training noise could be slightly deceptive; 3 out of our 4 VESPER_V20X iterations outperform every VESPER_V20 iteration in the Glacier Updates category. The 4th VESPER_V20X iteration is somewhat anomalous - the increased network complexity could mean that the model did not converge well for that particular iteration, for the glacier grid points. Since the updated V20 glacier fields are generally accurate globally, we saw no particular improvement in using VESPER_V15X to within the training noise. This suggests that the additional monthly lake maps are only useful if the underlying representation of static water is sufficiently accurate. Considering the particular glacier grid points we discussed previously in Section 3.1.3, the additional monthly lake maps were particularly useful for the Calluqueo glacier, with $\delta_{\text{V20X}} = 0.32$ compared to $\delta_{\text{V20}} = 2.49$ ($\sigma_{\text{V20}} = 1.59$, $\sigma_{\text{V20X}} = 0.73$). However we saw no improvement to within the training noise for the Juncal glacier

### 3.2.4 Timeseries

Thus far we have been focusing mainly on the $\delta_{\mathrm{VM}}$ metric averaged over the entire year of the test set. It is also of interest to explore how the prediction error for each of the 3 models varies with time. This is demonstrated in Fig 15 for each of the 4 updated categories that we have discussed.

For the Lake Updates and Lake-Ground Updates categories we can see that all the model predictions track the same general profile, with the error peaking in the northern hemisphere summer months. This is a result of FLake modelling being least accurate during the summer as the lake is not fully mixed and so the mixed layer depth for lakes is too shallow, resulting in skin temperatures with larger errors. Conversely, in the autumn and spring the lake is fully mixed and predictions have the smallest errors compared with observations. A clear hierarchy of models is evident; the VESPER_V20 and VESPER_V20X models consistently outperform VESPER_V15 across the year. This again is strong evidence, highlighted by VESPER, of the value of the updated fields both static and seasonally varying. We discussed previously how the annually and globally averaged $\delta_{\mathrm{VM}}$ values for the Lake Updates category were highly comparable for VESPER_V20 and VESPER_V20X. We can see from the top panel in Figure 15 that this equivalence is not consistent over the year. Instead, during the winter months of the northern hemisphere VESPER_V20 and VESPER_V20X are fairly equivalent; VESPER_V20 tends to outperform VESPER_V20X, but the difference is within the model training noise. However in the central months of the year VESPER_V20X starts to be slightly more accurate. This is likely for two reasons. Firstly, the monthly lake maps are in fact a climatology and therefore insufficiently precise to detect the exact ice on/off dates during the winter months, where we have a large number of grid points at high latitudes which will be subject to freezing, nullifying any time variability. The second reason is due to the accuracy of the lake mean depth which strongly drives the ice on-dates due to its influence on the heat capacity of the lake. During the warmer months lakes thaw, the monthly maps are more accurate, as the thawing of lake ice is mainly connected to the atmospheric conditions, not the lake depth, and so the information contained in them can be used to make more accurate predictions.

The Lake-Ground Updates timeseries broadly follows the same general profile as Lake Updates, but the errors are larger - those grid points where the lakes have been replaced with bare ground were particularly poorly described in V15. Additionally, for Lake Updates we see two sharp decreases in the prediction error during $\sim$ April and September, which are not as strongly reflected in Lake-Ground. This is due to the geographic location of the grid points in each of the two categories; for the Lake Updates category the grid points are located primarily in the boreal zones and so are subject to freezing and thawing over the course of the year leading to a strong seasonality due to the lake mixing that we have discussed. The sharp drop in April corresponds to a time where the lakes are unfrozen and fully mixed. However the lakes in the Lake-Ground sub-category are less concentrated and much more evenly distributed over the globe and so do not exhibit such a strong seasonality. Consistent with our previous discussion, the training noise makes it difficult to separate the predictions of the VESPER model for the vegetation category across the year. All generations of VESPER_VM follow the same general trend, with errors maximal at

the start and end of the year, and minimal during the spring and autumn months.

For the Glacier updates category, in order to deal with the separate warming and cooling seasonal cycles over the year, we separate grid points into the northern and southern hemispheres. For the northern hemisphere the errors peak for all models in the summer, again due to the lakes not being fully mixed. There is also an uptick in the prediction error for all models during the winter when the freezing is greatest - this indicates how ice cover can strongly influence the LST response. The familiar hierarchy of models is recovered; VESPER_V15 is generally outperformed by the more updated models. In turn VESPER_V20X is a general improvement over VESPER_V20 throughout the year, especially during the winter months where the training noise is minimal. Since this is the time when freezing is greatest, this suggests that the additional monthly maps and salt lake maps are particularly useful during this time. For the southern hemisphere the story is different. The errors are smallest during the middle of the year when we expect the freezing to be greatest. During the spring and autumn the errors are largest - this is correlated with a decrease in the number of observations suggesting that this is due to poorer data quality due to cloud cover. In the summer when the weather is clearer the errors start to decrease again. Given this variation in the data quality due to cloud cover it is difficult to draw any strong conclusions, and again for stronger performance cloud independent data should be used. What is obvious for the southern hemisphere glacier grid points is that the VESPER_V20 and VESPER_V20X models struggle to improve on VESPER_V15, unlike in the northern hemisphere. This suggests that the updated V20 fields are still insufficiently accurate for southern latitudes.

We have also discussed previously particular grid points that will likely show a large degree of temporal variability, or the lakes are saline, and as a consequence the static physiographic V15/V20 fields struggle to make accurate predictions (e.g. Table 5). In Figure 16 we present timeseries for two of these points: the Great Salt Lake Desert, Utah and Chott Felrhir, Algeria. Both these points were discussed in Sections 3.1.1, 3.2.1. We can see that for these two selected points the hierarchy of models no longer holds. Whilst there is a large degree of variability, and there is no clear separation between models for some parts of the year, generally it can be seen that VESPER_V20 performs worse than VESPER_V15. For the Great Salt Lake the inaccuracy when using the V20 physiographic fields is most pronounced during the summer months. April, May and June are some of the wettest months in this region. But the updated V20 fields specify a much smaller lake fraction than in V15 ($\sim 0.5$ compared to 0.0). Consequently during this time the V20 fields are maximally inaccurate and the prediction error of the VESPER_V20 model grows accordingly.This indicates again that the updated V20 fields are in fact over-corrections for this area. The inclusion of monthly lake maps and salt lake maps in VESPER_V20X notably reduces the error during these summer months. For Algeria, we can we can see that VESPER_V20 underperforms VESPER_V15 throughout the entire year. For this grid point the lake was completely removed when updating the V20 fields, with the lake fraction reducing from $\sim 0.35$ to 0.0. This also appears to have been an over-correction. The separation between the models is most pronounced in the early months of the year; in the winter months both the prediction error and the variance increase - this period is the wet season in Algeria where the wadi which feed Chott Felrhir fill up. Similar to the Great Salt Lake Desert, the inclusion of the monthly lake maps in VESPER_V20X improves the prediction accuracy, most notably in the early months of the year. Again, later in the year

the training noise is much greater and so it is harder to separate the predictions of the model, but on average VESPER_V20X outperforms VESPER_V20 over the entire year, highlighting the value of these additional physiographic fields. monthly fields.

## 4   Discussion

We have seen how VESPER can quantitatively evaluate the value of updates to the physiographic fields used by land-surface scheme, as well as identifying areas where the updates are inaccurate. For the former VESPER was able to show that the major regions where the lake surface parametrisation fields were updated - such as the Aral sea - enjoyed more accurate predictions, which verifies both the accuracy of the fields and their information content with respect to predicting skin temperatures. For the latter VESPER was able to identify grid points where the predictions became worse with the updated fields, indicating that the updated fields were in fact less accurate. More generally we have also seen how detailed knowledge of surface water fields (e.g. up to date permanent water distribution, seasonal water distribution, salt lake distribution, etc.) can notably improve the accuracy with which the skin temperature can be modelled, e.g. grid points with significant updates (i.e. where the field has changed by $\geq 10$ %) to the lake fields show a mean absolute error reduction of skin temperature globally of 0.37K (Table 4). Given the performance of VESPER it may be possible in the future to update or correct the input fields at a high cadence, e.g. yearly or even more frequently.

There are multiple possible further extensions of this work. We have not currently included the errors on the MODIS observations into the VESPER model. During the "matching-in-space" step relating the ERA and MODIS data (Section 2.2), it could be a worthwhile extension to weight the averaged MODIS points by their corresponding errors (e.g. Fig. 5b) when deriving a single MODIS observation for a given ERA grid point. This would then provide a more accurate and confident representation of the true surface temperature at a particular space-time point. Due to the inherent stochasticity of training a model we have seen that some grid points have a particularly large training noise. To better quantify this effect and try to draw stronger conclusions for this subset of points it would also be desirable to train an ensemble of models ("ensemble learning") and combine the predictions from multiple models to reduce this variance. Additionally, our examination of the value of the monthly lake maps is only a preliminary study. It would be of interest to follow seasonal lakes over a longer time period (e.g. decadal) beyond the 12 month maps that we use, in order to better quantify their time variability, as well as the differences between years (e.g. if the lake fraction was particularly high in the January of one year, but low in the subsequent year). It would also be of interest to try to quantify if VESPER and ECLand respond to changes in the input physiographic fields used by land-surface scheme in the same way, which is key to be able to then apply the VESPER results to the full earth system model development. Since VESPER is trained on ERA5, if we want to model the outputs of the IFS we must assume that the statistical behaviour of the input fields does not change from ERA to IFS. This is a fair assumption, but it would be interesting to investigate this quantitively in greater detail. We have focused here primarily on hydrological applications, our primary concern being the ability to evaluate the parametrised water body representation, however the general application of the method for any updated fields that we

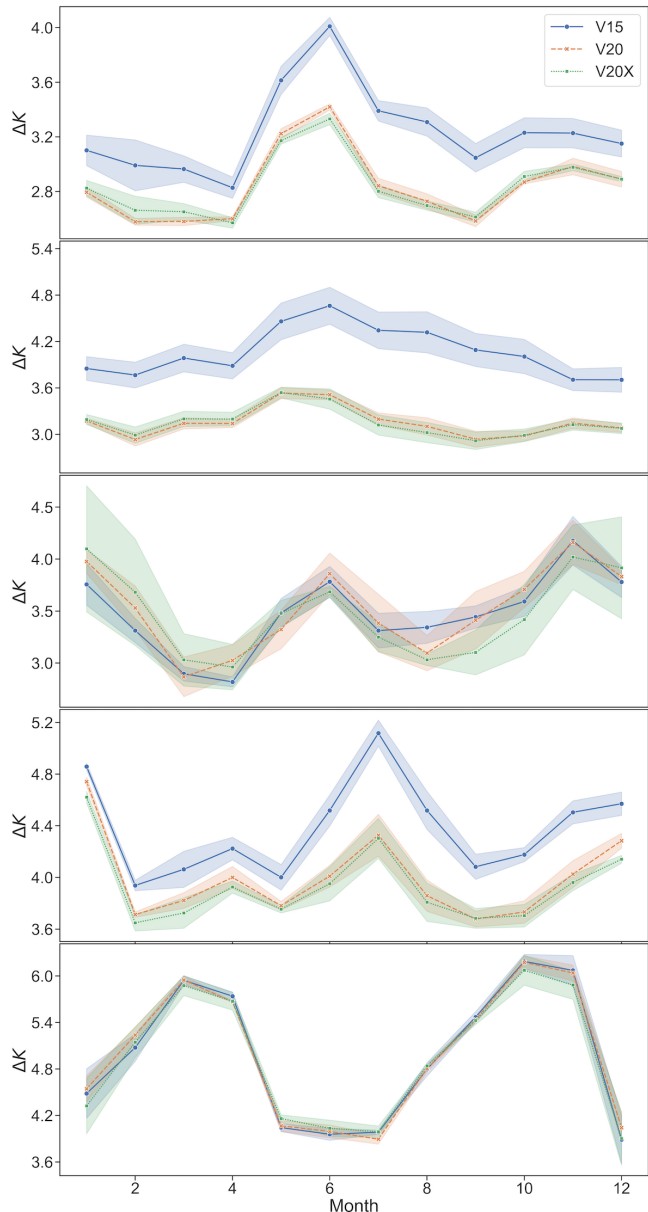

**Figure 15.** Mean prediction error in the surface temperature $\Delta K$, averaged over all grid points, for each of the 3 models over the course of the test year for (*top panel*) Lake Updates, (*second panel*) Lake-Ground Updates, (*third panel*) Vegetation Updates, (*fourth panel*) Glacier Updates, northern hemisphere and (*bottom panel*) Glacier Updates, southern hemisphere. The shaded regions show the $1\sigma$ training noises. For the Lake categories, all models follow the same general profile, with the VESPER_V20 and VESPER_V20X models generally outperforming VESPER_V15 model over the year.

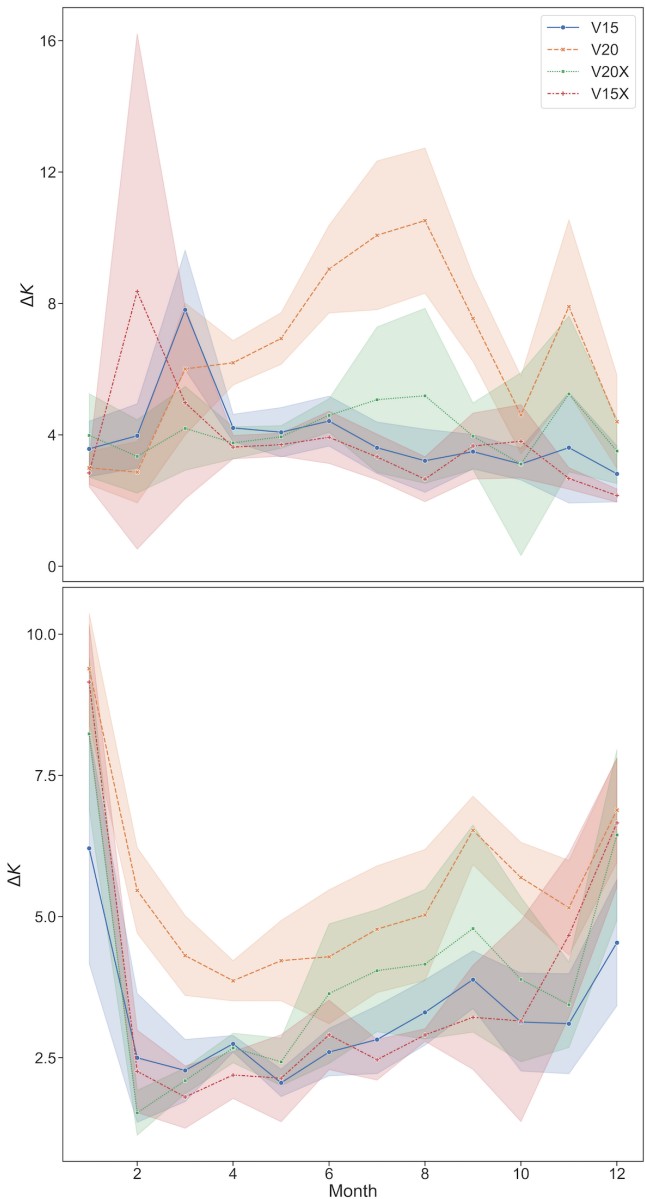

**Figure 16.** Variation in the prediction error for the grid points at Great Salt Lake, Utah (top panel) and Chott Felrhir, Algeria (bottom panel). There is a large degree of variability, but for both grid points VESPER_V20 model is generally less performant than VESPER_V15 , indicating that the updated V20 fields are less accurate here. Corrections introduced by the augmented VESPER_V20X model with saline and monthly lake maps outperform those without, indicating the value of these fields in these regions.The shaded regions show the $1\sigma$ training noises.

want to assess could also be explored. Extension to non-lake hydrological fields like wetland extent or river bathymetry model parameters, or even non hydrological fields such as orography would be an interesting further development. The development of a more mature, integrated pipeline for automatically evaluating updated parametrisations could also be a worthwhile pursuit.

Another natural and interesting extension of this work would be to use VESPER to perform a feature importance or sensitivity analysis for the various input fields of the neural network. Additionally, an approach which may prove fruitful in the enterprise for improved parametrised representation of water bodies is to invert the problem and treat VESPER as a function to optimise. That is to say, VESPER can be thought of as a function which takes some inputs - in this case a lake parametrisation - and returns a loss metric i.e. how accurate the predictions are compared to the test set. Given this loss metric it may then be possible to vary the inputs and use standard optimisation techniques to learn the optimal parametrisation. Whilst this may be an expensive technique as there are effectively two nested models over which to optimise (for every optimisation step in the higher model, one must train the VESPER network from scratch) it could be possible given appropriate hardware or with reduced data focusing just on targeted locations (e.g. "What is the best way to represent the lakes in this area?"). The loss gradient information can also be used to tune individual features, informing whether an input variable should be larger or smaller.

## 5 Conclusion

Weather and climate modelling relies on accurate, up-to-date descriptions of surface fields, such as inland water, so as to provide appropriate boundary conditions for the numerical evolution. Lakes can significantly influence both weather and climate, but sufficiently accurate representation of lakes is challenging and the natural changes in water bodies mean that these representations need to be frequently updated. A new method based on a neural network regressor for automatically and quickly verifying the updated lake fields - VESPER - has been presented in this work. This tool has been deployed to verify the recent updates to the FLake parametrisation, which include additional datasets such as the GSWE and updated methods for determining the lake depth from GLDBv3. The updated parametrisation fields were shown globally to be an improvement over the original fields; for a subset of grid points which have had significant updates to the lake fields, the prediction error in the skin temperature decreased by a MAE of 0.37K. Conversely, VESPER also identified individual grid points where the updated lake fields were less accurate, enabling these points to subsequently be corrected, such as incorrect removal of lake water and losing forests to bare ground. Multiple further extensions of this work, including extension to non lake fields and the development of a more mature integrated pipeline have been discussed.

*Code availability.* The code used in constructing VESPER, including the methods for joining the ERA and MODIS datasets and the construction of the neural network regression model is open-sourced at https://github.com/tomkimpson/ML4L

*Author contributions.* All the authors contributed equally to the work.Tom Kimpson wrote the manuscript with contributions from all other authors.

*Competing interests.* The authors declare that they have no conflict of interest.

*Acknowledgements.* This project has received funding from the European Research Council (ERC) under the European Union's Horizon 2020 research and innovation programme (Grant No 741112). PD gratefully acknowledges funding from the ESiWACE project funded 725 under Horizon 2020 No. 823988. PD and MC gratefully acknowledge funding from the MAELSTROM EuroHPC-JU project (JU) under No 955513. The JU receives support from the European Union's Horizon research and innovation programme and United Kingdom, Germany, Italy, Luxembourg, Switzerland, and Norway.

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
