# Peer review of "Deep learning for quality control of surface physiographic fields using satellite Earth observations"

_EGUsphere, 2022_

## Referee Comment (RC1)

**Dear authors,**

In the following, there are several important essential comments and some editorial comments.

**Essential comment #1: mixing of different models, parameters and predictors**

In the paper, you mix a physical model and a neural network regression model, which leads to wrong statements, wrong-posed questions and wrong conclusions. A physical model in your study was IFS (with FLake and HTESSEL included). Lake cover fields are **parameters** for it. A neural network model*s* in your study are several versions of VESPER (several sets of VESPER's internal coefficients) trained for different sets of predictors. Sets of lake cover fields are **predictors** for these VESPERs. This is an important difference between IFS/HTESSEL/FLake and VESPER, how they treat lake fields: are they parameters or predictors. Also, you state that you mix the physical lake parameterisation itself (FLake model) and model parameters (L54-55). This overall mix leads you to wrong conclusions. Starting from the very beginning:

L.79, L82-87: *"– Are the updated fields informative?", "For instance, the main target of lake parametrization is to reproduce lake surface water temperatures (and therefore evaporation rates). If a lake parametrisation scheme is updated to better represent different types of inland waterbodies, the time variability of inland waterbodies and/or the lake morphology fields use more in situ measurements, does this additional information allow for more accurate predictions of the lake surface water temperatures? Is it therefore worthwhile to update the parametrisation in this way?"*. Yes, additional **accurate** information about lake cover fraction and lake depth should **always** allow FLake to improve simulation of fluxes and water surface temperatures over the appropriate part of the IFS's grid box! Additional **accurate** information about vegetation should also **always** allow better simulations for HTESSEL. Just because they are deterministic physical models. Of course provided, that these deterministic physical models describe physical processes correctly and there are no compensating errors (which is usually not the case, but this point is actually out of scopes of your study, this is usually treated by data assimilation). However, for a neural network regression model, the situation is different! Improving of information about lake cover fraction also will **always** lead to improving of VESPER simulations, but only **in general**! Local degradation may happen! This is an essential feature of any statistical model: one particular prediction may be very bad, but general statistics is always good. The particular simulation result might not depend on a quality of predictor(s); it just depends on a model, its internal nature. And this is what you have in your study. Thus, the question "Are the updated fields informative" is just wrong-posed, both for a physical model and a neural network model.

2.2. L88-89: *"This problem of quickly and automatically verifying the accuracy and information gain of updated lake parametrisations is the aim of this work."* - this is wrong-posed task. You can't verify information gain. You can only verify the modeling results and compare different models. And in your study you are actually comparing verification scores of different versions of VESPER, when it is trained with different sets of predictors. Yes, it has some relation also for the physical model simulations. But it is not the same!

2.3. L105-106: *"In Section 3 we then deploy VESPER to investigate and evaluate updated lake parametrisation fields."* No, you compare the results of different versions of VESPER in terms of the local impact of new predictors (new lake cover and salinity flag). Again, it has some relation to physical model simulations, but it is not the same!

2.4. L19: *"The neural network regression model has proven to be useful and easily adaptable to assess unforeseen impacts of ancillary datasets"* - too optimistic! You may just make a suggestion about impacts, because you consider different models.

… and so on, in the whole text! I don't continue with examples. They are on every page. As a result, the style is too optimistic, discussions are messy and mostly speculative (see also Essential comment #2). The whole discussion goes as if updates were made to the physical model, not to VESPER! **The text should be re-thought and totally rewritten, including Discussion and Conclusions.**

**Essential comment #2: strange results and speculative explanations.**

**Lake freezing and results for Northern Canada:** Lake ice freezing and melting is not reflected in the lake fraction fields! When a lake freeze, it does not turn to soil. Also vise versa, lake fraction does not affect freezing and melting! For the physical model, changing of lake fraction can't explain increasing of errors. There is nothing specific for FLake over Northwest Territories. For FLake, but not for VESPER. From the point of view of physics, presence of ice can't explain deterioration or improvement of predictions between V20 and V15 specifically for Northwest Territories (L375). And especially between V20 and V20X (L498-508), because lakes in Canada are very stable and not saline. The reason is something else. Perhaps, it is just statistical nature of VESPER: it was re-trained with different predictors. The same about related parts in Section 3.2.4.

**Small changes in predictors lead to large changes in the results:** the mean monthly lake fraction correction of 0.1% for the Toshka lakes leaded to changing of the VESPER results (V20 vs V20X) in 2.15 K (L482). This again seems to be a characteristic feature of the statistical model. Such change would never happen in a physical model! It would ask for a difference of 2000 K between the lake water surface temperature and bare soil temperature in the same grid box! For other lakes, much larger changes in the lake fraction leaded to approximately the same corrections (L480-495). Again, I can suggest that this is a characteristic feature of VESPER itself, rather than *"a nice confirmation to have these updated fields verified by VESPER"* (L493).

**Vegetation updates:** Sections 3.1.2 and 3.2.2. In the V20 experiment, you are tried to find the reasons of the bad VESPER performance in the wrong predictors (vegetation fraction). And really, you found problems. But in V20X experiment, almost no predictor values were changed in terms of physics: vegetation fraction was not changed and was still wrong, lake fields were changed only slightly, no changes at all at Fig. 7. And discussing the results of this experiment, you refer large improvements in the performance to very small changes in the lake cover fields! In physical model, this would never happen. But here in VESPER, the real reason (to my mind) is that it was re-trained. You just applied the new model with different predictors (for vegetation points – formally, but not physically), and the new model gave better results, in this case also locally. The same about related parts in Section 3.2.4.

**Glacier updates in V20X:** Section 3.2.3. Freezing and melting of lakes is not relevant to changing of lake fraction; there are no saline or ephemeral lakes in the vicinity of glaciers. That is why, for the physical model one can't expect much improvements of the skin temperature fields in glacier points with implementation of changing lake cover and salinity. However, with VESPER this happened! Again, with almost no local changes in predictors in terms of physics (only formal change), suddenly there was an improvement of VESPER performance between V20 and V20X. And again, this is perhaps related to the fact that VESPER was re-trained. It does not demonstrate the importance of lake salinity information close to glaciers. As for vegetation points, you just appled the new model with different predictors, and the new model gave better results, in this case also locally. The same about related parts in Section 3.2.4.

**Essential comment #3: MODIS observations**

In your study, you considered MODIS observations as a ground truth. However, they also contain errors. Sometimes errors are very large, especially for lakes, mainly due to miss-classification between cloud water and lake water. MODIS errors may explain partly your strange results (see Essential comment #2). You calculate a map of errors from information provided by MODIS developers (Fig. 2), but these are just errors of MODIS algorithms, not obtained by comparing with in-situ observations or so. I would advise you to look into data itself. Are they realistic, especially for your strange situations? By the way, Fig. 2 is located at Page 8, but you refer to it only at Page 30.

**Essential comment #4: Errors.**

Everywhere in the paper, the LST errors are positive. It is not specified what kind of errors are they, just "prediction error", "prediction accuracy" or so. Obviously this is not just a mean difference between the observed and predicted values (bias), because bias can be both positive and negative. What is it, then? How it was calculated? Is it a mean absolute error? Or root mean square error? Or different errors in different parts of the paper? Please specify! By the way, errors are very large locally, up to 17K (Fig. 10) … Was the predicted LST higher or lower then observed in these cases?

**Essential comment #5: training and evaluating periods.**

In your study, the training and evaluating periods were 1 year, namely 2016 for training and 2019 for evaluating. However, if you repeat the exercise for different years, you might have different results, perhaps even opposite! Even for Laka Eyre in Australia: it has a full coverage several times per century, but floods with 1.5-4 m depth occur once in 3-10 years. If you choose a year when Lake Eyre contains more water, you might get opposite results: deterioration of the forecast with new data, where the lake in removed. For seasonal lakes, in order to follow properly their annual evolution, you need 30-years (at least 10-years) study. If you have no technical possibility for that, you can only preliminary suggest that e.g. the monthly data on the lake coverage may give some positive impact. Also, you can't consider any annual cycle of errors. Thus, the whole Section 3.2.4 is questionable. I even don't comment it much. Please elaborate.

**Essential comment #6: VESPER does not "beat" ERA5, it corrects ERA5.**

You included the ERA5 skin temperature in your vector of predictors for VESPER (Table 1 line 7), along with many other values: it means that VESPER will **always** produce something better then just the ERA5 skin temperature itself. Again, this is for general statistics, not necessarily locally. With this approach you just correct the ERA5 skin temperature, to have better general performance. This is why you can't say, e.g., that *"the trained model generally enjoys increased accuracy over the ERA5 predictions"* (L207-206) or *"… It can be seen that the network generally outperforms the ERA5 predictions … In contrast the network demonstrates generally good performance"* (capture to Fig.4). You can only say that VESPER corrects ERA5 successfully. By the way, it is interesting to know, which part of dispersion in the statistical model goes back to the corrected value (skin temperature) itself and to the very closely related 2 meter temperature and relative humidity (Table 2 lines 5-6); and which part to other parameters. Also, you can't say that (L212-214) *"More fundamentally, this also indicates that there is some information captured in the input fields to the network that is not expressed through the current ERA5 reanalysis modelling."* - this is just a trash. There is no additional information in ERA5 fields on top of ERA5 fields. Correction by VESPER is achieved first of all through the use of ***observations*** for training.

**Essential comment #7: technical names.**

Please remove code variable names, technical specifications, etc. and give mathematical or physical descriptions instead (when relevant). Examples are:
L177-178: *"Step 2 in the joining pipeline uses a GPU-accelerated k-nearest neighbours algorithm (RAPIDS, v22.04.00)"*
L243-244: *"… with coarse resolution aggregation technique MEAN … with coarse resolution aggregation technique MODE ..."*
Table 1, column 1 – to remove.

**Essential comment #8: predictors.**

Do I understand correctly, that for VESPER V20,V20X and V15X you just added the new fields into the vector of predictors? You did not calculate differences between "old" and "new" fields? If yes, please say it clearly, without using euphemism "corrections". If not, please provide more details: how you treated inconsistencies then? Also, please add units in Table 1. What is "Ice temperature layer 1,2" in this table? Is it a sea ice temperature? What is "Geopotential"? Is it orography times gravitation acceleration or some other geopotential? Please add to Table 1 also predictors for the experiments V20X/V15X: changing lake fractions and salinity, also with units.

**Essential comment #9: vegetation and glacier updates.**

How it happened, that due to updating of lake information only, also vegetation and glacier fields were changed? Why forests were removed? Explanation at L399 is insufficient. What happened with glaciers? Where they removed or added? Why? Please provide more information.

**Essential comment #10: significant figures.**

Since the temperature is measured with the accuracy of 0.1K, prediction error statistics can't have the accuracy higher than 0.01K (can't contain more that 2 significant figures). For example, in Table 2, the value -1.12 is correct, but the value +0.0048 is incorrect, should be rounded to +0.00. Please elaborate, in Table 2 and throughout the whole text. This will also influence some of your conclusions.

**Essential comment #11: seasonal lake fraction changes and salinity.**

Experiments V20X and V15X were designed so that the monthly lake fraction fields and salinity fields were added to VESPER predictors together. This does not allow to distinguish between the impact of these two factors. Often, the coverage of saline lakes really changes throughout the year, but it is not always the case. Lakes with low salinity are usually stable. To considering (i) varying lake fraction and (ii) lake salinity in IFS would need very different research and developments. So, it would be interesting to see if these two factors separately can improve VESPER corrections.

**Essential comment #12:** What are shadows on Fig. 9-10? Please explain.

**Some editorial comments:**

1. L5: *"... the lower boundary conditions such as … and moisture availability near the surface"*. Moisture availability can't be a lower boundary condition; only a flux or a physical value can be.
2. L24 *"Their distribution is highly anisotropic ..."*. Anisotropy is a statistical property of a field to have different spatial correlations in different directions (e.g. to the North and to the East). Here, I guess, you mean something more simple: just non-uniform.
3. L111. *"The target is the empirical land surface temperature (skin temperature)."* The target of simulation can't be empirical. Please reformulate.
4. L219: *".. used in the IFS to generate the lake parameters is the GlobCover2009 global map"* => used in the IFS to generate the lake fraction is the GlobCover2009 global map
5. L245-246: *"GLDBv3 increases the total number of lakes with in situ information by ∼ 1500 , introduces a depth distinction between freshwater and saline lakes, and updates the method by which the lake depth is calculated based on climate type and geological origins."* => GLDBv3 increases the total number of lakes with in situ lake depth information by ∼ 1500 , introduces distinction between freshwater and saline lakes (this information is currently not used by FLake), and suggests the method to assess the depth for lakes without in-situ observations using geological and climate type information.
6. L248-249: *"Verification of the updated lake depth fields against 353 lakes in Finland shows that GLDBv3 exhibits a 52%  bias reduction compared to GLDBv1 (Choulga et al., 2019)."* - bias in which field? I the lake depth? Or in simulated temperatures?

**Editorial, impossible to understand or too long:**

1. L74-76: "It is however challenging to accurately represent lakes in these parametrisations; the majority of lakes which are resolved at a 9km grid spacing have not had their morphology accurately measured, let alone monitored, whilst 28.9% of land and coastal cells are treated for sub-grid water."
2. L95-99: "Indeed, the early successes of these machine learning methods have led to the development of a 10-year roadmap for machine learning at ECMWF (Düben et al., 2021), with machine learning methods looking to be integrated into the operational workflow and machine learning demands considered in the procurement of HPC facilities; the ongoing development of novel computer architectures (e.g. GPU, IPU, FGPA) motivates utilizing algorithms and techniques which can efficiently take advantage of these new chips and gain significant performance returns."

**Noticed typos:**

1. L33, L736: "lu" => Lu.
2. L144: "… which takes a spherical projection ellipsoid but a WGS84 datum ellipsoid ..." => which takes a spherical projection but a WGS84 datum ellipsoid
3. L185-186: should it be a bar over *y*?
4. Table 1: "How vegetation cover" => "High vegetation cover"
5. Capture to Figure 5: "7 points" => 6 points

---

## Author Comment (AC1)

Response to Reviewer 1 Comments concerning HESS submission egusphere-2022-1177

Tom Kimpson      Margarita Choulga      Matthew Chantry

Gianpaolo Balsamo      Souhail Boussetta      Peter Dueben      Tim Palmer

June 1, 2023

Author response to referee reports for the paper egusphere-2022-1177, entitled *"Deep Learning for Verification of Earth-System Parametrisation of Water Bodies"*.

We thank the referee for their reading, helpful criticism and suggestions towards the improvement of the manuscript.

We have addressed each point in turn below and the manuscript has been updated accordingly

We hope that this satisfies the request for changes necessary to proceed with the publication of the updated manuscript.

**Essential comments**

1. **Essential comment 1: mixing of different models, parameters and predictors**

   We agree with the general points here raised by the reviewer that the terminology used was insufficiently precise. This has now been corrected throughout the entire manuscript. We now make it clear that we are always comparing the results between two neural network models, not comparing NN predictions with e.g. FLake. If we update some input field to our NN and the NN prediction accuracy improves, this is good evidence that the updated field itself is more accurate and would enable a physical model like FLake to make more accurate predictions. We note that this work is a first attempt / investigation to explore the possibility of using these kind of machine learning methods for a global check of surface physiographic fields, and we will look to refine and further develop these methods in future.

   Taking some individual points:

   - **L.79, L82-87**. What was meant here is the distinction between is the new information informative enough to have visible changes, or it is just lots of work for no impact? This has now been properly phrased in the text. Whilst we did briefly discuss local degradation previously we have also now included a more explicit discussion. Additionally, we explicitly calculate the training noise by retraining VESPER multiple times. We now highlight the effect of the training noise in the updated manuscript. Having retrained the model multiple times, our conclusions are generally unchanged for all grid-point categories, with the exception of the Vegetation category which has significant training noise over a small number of grid cells making it difficult to draw meaningful conclusions. This is again discussed in the text and we thank the reviewer for raising this point for our attention.

   - **2.2 L88-89** As the point above, we have retrained VESPER multiple times to show that the training noise is generally smaller than the prediction changes due to the different input fields. We reword the text throughout to be clear that we are always comparing two NN models.

- **2.3 L105-106** As above, terminology and text has been changed throughout the manuscript to be more precise
- **2.4 L19** As discussed we have now trained multiple versions of the model to better quantify the training noise and our conclusions remain unchanged that we can use VESPER to (a) check that an updated field is closer to reality and (b) see if this updated field increases the accuracy of our NN model. Both of these points are relevant for the updated fields within a physical model like FLake.

2. **Essential comment 2: strange results and speculative explanations**

As mentioned we now explicitly calculate the training noise by retraining VESPER multiple times. For the points in Northern Canada, the Toshka lakes and the Vegetation category our previously quoted changes are less than the training noise. Again we thank the reviewer for highlighting this to us. We have removed the discussion on Northern Canada and the Toshka lakes from the manuscript, whilst the effect of the noise on the Vegetation category is now discussed explicitly in the text. We emphasize that our main conclusion re the lake and glacier categories are unchanged by this retraining - the difference in the improvement due to the updated fields is much greater than the training noise.

3. **Essential comment 3: MODIS observations**

A discussion on the quality of MODIS data has been added to the text.

4. **Essential comment 4: Errors**

Throughout the work we use an absolute error i.e $|\text{LST predicted by VESPER} - \text{LST from MODIS}|$. This is now specified explicitly in the text. We have also explored the use of different error metric such as bias and RMSE, but our conclusions remain unchanged.

5. **Essential comment 5: training and evaluating periods**

We have trained VESPER with different input years (i.e. 2018, 2019) and results were the same. For monthly data training we agree that more data is needed and this work can be considered as our first attempt to represent and evaluate monthly lakes - the updated lake fields themselves are only for a single 12 month period. We have reworded the text accordingly to make this concession that this is our first attempt to include monthly lakes. Additionally data preparation should be more detailed - it would be useful to consider only grid cells with constant cover over the training period.

6. **Essential comment 6: VESPER does not "beat" ERA5, it corrects ERA5**

Fully agree, corrected in text accordingly

7. **Essential comment 7: technical names**

We have updated the definitions of the aggregation techniques in the text. For the $k$-nearest neighbours algorithm we feel that this is a sufficiently well known technique, common in many ML texts that it is sufficient to specify the technique used and reference the specific Python library (RAPIDS) that we used.

8. **Essential comment 8: predictors**

The different VESPER models do indeed have a different vector of predictors. This has now been specified explicitly in the updated manuscript, along with better definitions with units of the various input fields (see Tables 1-3)

9. **Essential comment 9: vegetation and glacier updates**

The vegetation and glacier fields were updated in proportion to the change in the lake fraction. For instance if before the fraction of the cell which is lake $= 0.75$ and the fraction which is glacier $= 0.25$, and then after the update the fraction of the cell which is lake $= 0.80$, the new glacier field is 0.20. This is now discussed in the updated manuscript.

10. **Essential comment 10: significant figures**

   Agreed. Corrected throughout manuscript

11. **Essential comment 11: seasonal lake fraction changes and salinity.**

   This is an good point. For this work we are satisfied to consider the combined affect of monthly maps and salt lakes, since many of the locations we specifically highlight in the manuscript are saline lakes with large expected time variability in the surface water. Further work in this area is currently ongoing and we defer a more in depth, global study of saline lakes and monthly maps for the future.

12. **Essential comment 12: What are shadows on Fig. 9-10? Please explain.**

   Previously these shadows were confidence intervals. In the updated manuscript, with multiple VESPER trainings these shadows are the $\pm 1\sigma$ bounds. We have specified this in the Figure captions.

**Other comments**

All typos and editorial comments have been corrected in the updated manuscript.

---

## Author Comment (AC2)

Response to Reviewer 2 Comments concerning HESS submission egusphere-2022-1177

Tom Kimpson       Margarita Choulga       Matthew Chantry

Gianpaolo Balsamo       Souhail Boussetta       Peter Dueben       Tim Palmer

June 1, 2023

Author response to referee reports for the paper egusphere-2022-1177, entitled *"Deep Learning for Verification of Earth-System Parametrisation of Water Bodies"*.

We thank the referee for their reading, helpful criticism and suggestions towards the improvement of the manuscript.

We have addressed each point in turn below and the manuscript has been updated accordingly

We hope that this satisfies the request for changes necessary to proceed with the publication of the updated manuscript.

**Comments**

The reviewer suggests a major overhaul of the structure of the manuscript. We agree that this is a very good suggestion and the text has been completely restructured as recommend. We now present the construction of VESPER much more thoroughly, detail the various input fields, and specify the differences between the various VESPER generations. Only then do we then go on to deploy VESPER on lake fields and discuss the results. Where possible we have made an effort to be more concise.

- **l.54** The terminology re parameters, model and physiography has been updated throughout the text. The tables have also been updated to describe the choice of variables, and the different VESPER models (see e.g. Tables 1-3 in update manuscript).

- **l. 123 ERA5** All required information has been added to the text.

- **l. 133 MODIS** All required information has been added to the text.

- **l. 146 and Figure 3** By 4km resolution, we were referring to the resolution at the equator. This has now been updated in the text. Re the number of points at high latitudes, this is a natural consequence of the MODIS orbit, see e.g. animation at `https://svs.gsfc.nasa.gov/3348`

- **l. 184** This change has been made and a new table (Table 3) added which specified each VESPER configuration.

- **Figure 4 and related text**

  The prediction error is now defined at the end of section 2.4. Whilst the performance of VESPER relative to ERA5 is encouraging, one aspect of this is that VESPER has been trained directly on MODIS data whereas ERA5 has not. For this work we take MODIS data as our source of truth - as far as VESPER is concerned the MODIS data is reality, whereas of course the MODIS data has its own errors and systematics. The question of if a deep

learning model could be used for forecasting is a very interesting one, but slightly beyond the scope of this study - we are primarily interested in quickly evaluating the accuracy of the fields that get passed to a dynamical model.

- **Section 3 Results**

  This section has been restructured to just contain the lake results, rather than the VESPER configuration as requested.

  A short discussion on how the non-lake climate fields such as vegetation cover or orography are update in response to the update in the lake fields is not included at the end of Section 2.2.1 c.f. Aral sea. We have tried to condense this section, but generally there is lots to discuss and we do prefer to be thorough here. As suggested there is plenty of further discussion to be had on e.g. monthly lake maps, glaciers etc. but we defer this for a future study

- **Section 4 Discussion**

  We have added the referee's point about if VESPER and ECLand parametrisations would react similarly to changes in the input fields to the discussion. In short, this is a very interesting question. The use of a tool that was trained from ERA5 in model output of IFS requires the assumption that the statistical behaviour of the fields does not change from ERA to IFS as the ML model would otherwise be forced to extrapolate (which it will not be able to do). This is a fair assumption, but it would be interesting to investigate this quantitively in greater detail. We defer the investigation of this question to a future work and thank the referee for raising a good point.

---

## Referee Report (RR1)

Reviewer comments on the second version of

Deep Learning for Verification of Earth-System Parametrisation of
Water Bodies by
Tom Kimpson, Margarita Choulga, Matthew Chantry, Gianpaolo Balsamo,
Souhail Boussetta, Peter Dueben and Tim Palmer

The manuscript has changed a lot and the presentation improved significantly. My
comments on the first round have been taken into account and the authors'
response was sufficient. However, there are still minor inaccuracies all around
the text, some of which I have spotted and mention in an annotated pdf of the
manuscript. I would recommend to check carefully the formulations in the
abstract, and also think about its relations to the conclusions. After minor
corrections the manuscript seems ready for publication.

[revised manuscript text omitted]

---

## Author Response (AR2)

Response to Reviewer Comments concerning HESS submission egusphere-2022-1177

Tom Kimpson        Margarita Choulga        Matthew Chantry

Gianpaolo Balsamo        Souhail Boussetta        Peter Dueben        Tim Palmer

October 21, 2023

Author response to referee reports for the paper egusphere-2022-1177, entitled *"Deep Learning for Verification of Earth-System Parametrisation of Water Bodies"*.

We thank the referee for their additional useful comments which we feel have notably improved the manuscript. All comments and suggestions have been implemented in the updated manuscript.

We have addressed each point in turn below.

We hope that this satisfies the request for changes necessary to proceed with publication

Reviewer comments on the second version of

Deep Learning for Verification of Earth-System Parametrisation of
Water Bodies by
Tom Kimpson, Margarita Choulga, Matthew Chantry, Gianpaolo Balsamo,
Souhail Boussetta, Peter Dueben and Tim Palmer

The manuscript has changed a lot and the presentation improved significantly. My
comments on the first round have been taken into account and the authors'
response was sufficient. However, there are still minor inaccuracies all around
the text, some of which I have spotted and mention in an annotated pdf of the
manuscript. I would recommend to check carefully the formulations in the
abstract, and also think about its relations to the conclusions. After minor
corrections the manuscript seems ready for publication.

[revised manuscript text omitted]